# Edge computing based english translation model using fuzzy semantic optimal control technique

**Na Wang** (ID) *

Jiaozuo Teachers College, Jiaozuo, Henan, China

* wangna188@outlook.com

## Abstract

People's need for English translation is gradually growing in the modern era of technological advancements, and a computer that can comprehend and interpret English is now more crucial than ever. Some issues, including ambiguity in English translation and improper word choice in translation techniques, must be addressed to enhance the quality of the English translation model and accuracy based on the corpus. Hence, an edge computing-based translation model (FSRL-P2O) is proposed to improve translation accuracy by using huge bilingual corpora, considering Fuzzy Semantic (FS) properties, and maximizing the translation output using optimal control techniques with the incorporation of Reinforcement Learning and Proximal Policy Optimisation (PPO) techniques. The corpus data is initially gathered, and necessary preprocessing and feature extraction techniques are made. The preprocessed sentences are given as input to the fuzzy semantic similarity phase, which aims to avoid uncertainties by measuring the semantic resemblance between two linguistic elements, such as phrases, words, or sentences involved in a translation using the Jaccard similarity coefficient. The fuzzy semantic resemblance component's training estimates the degree of overlap or similarity between two sentences, such as calculating the percentage of characters and length of the longest matching sequence of characters. The suggested Reinforcement learning and PPO can address specific uncertainty causes in machine translation assessment, like out-of-domain data and low-quality references. In addition to simple word-level comparison, it permits a more complex grasp of the semantic link. Reinforcement Learning (RL) and Proximal Policy Optimisation (PPO) techniques are implemented as optimal control techniques to optimize the translation procedures and enhance the quality and precision of generated translations. RL and PPO aim to improve a machine translation system's translation policy depending on a predetermined reward signal or quality parameter. The system's effectiveness is evaluated by various metrics such as accuracy, Fuzzy semantic similarity, Bi-Lingual Evaluation Understudy (BLEU), and National Institute of Standards and Technology score (NIST). Thus, the proposed system achieves

**Data availability statement:** All relevant data are within the manuscript and its Supporting information files.

**Funding:** This research was funded by 2021 Teacher education curriculum Reform research project of Henan Province: Research on the Application of Temperament Theory in the Teaching of Primary Education Major (Key Project) [2021-JSJYZD-04]. The annual general project of the 14th five-year Plan of Education science in Henan Province: Research on the Cultivation Mode of Primary Education Specialty from the Perspective of Temperament [2021YB0704]. Henan Education Department project: Research on the Application of Temperament Theory in Normal College Teaching [2019GZGG044]. The funders had no role in study design, data collection and analysis, decision to publish, or preparation of the manuscript.

**Competing interests:** The authors have declared that no competing interests exist.

higher quality and translation accuracy of the text that has been translated and produces higher semantic similarity.

---

## 1. Introduction

Corpus-based translation models use large corpora of parallel texts to identify similarities and translational equivalents. These corpora, which are used to recognize and learn translation patterns, are made up of texts in the associated original and intended languages, either at a sentence or phrase level, to enhance the effectiveness and precision of English translations [1]. The Internet business is growing quickly along with the economy, and English translation's standing in international trade is steadily rising. By reducing the cost and time of human translation, machine technology for translation can solve several issues with human translation [2]. The goal and challenge of early machine translation systems, which employed computers to translate between natural languages, was natural language processing study [3]. An artificial translation model for scoring is used based on backpropagation NN to improve English language teaching. The result improved the score's accuracy, but the study nonetheless has reference value despite the restrictions of the theoretical level [4]. Khan et al. [5] suggested translating sentences written in English into a similar Pakistani language regarding sign statements using a grammar-based machine translation algorithm. Accuracy improved with a BLEU score of 78%, and the result demonstrated that it successfully translates simple sentences and fails to translate compounded and compound lengthy phrases precisely. Yang et al. [6] integrated the encoder-decoder architecture using a fuzzy semantic representation (FSR) approach to identify the semantics of uncommon vocabulary using a hierarchy-based clustering system to group unusual words. The hyperparameters must be properly tuned and optimized to get the best results. By calculating the fuzzy semantic network's distance [7], the English semantics must be categorically arranged to find their ideal likeness. It produced the best possible translation outcomes, with a BLEU of 26.1 with an accuracy of 95%.

Wei recommended a system design using data mining to match intelligent semantic algorithms for collecting English-Chinese content in the library [8]. This algorithm recognizes input features that examine the parallel corpus's input language. The reliability of the outcomes could be impacted by the part-of-speech dictionary employed for comparison. Dong et al. examined the challenge of selecting subsets from enormous speech corpora by incorporating data contributions from time-continuous utterances [9]. These multi-tag constraints are not restricted to single-scale metrics. Statistical n-gram models are used with consistent coverage of the target and its internal attributes and improved accuracy. Ma and Di [10] suggested a semantic ontology translation paradigm that applied fuzzy mapping for translating data with proofreading and a function of decision-making. This method lowers the cost of human proofreading, increases English proofreading effectiveness and precision, and can accommodate user requests. The important problem is the translation memory's

frequent updating of semantics. The linguistic schools initially employed the rule-based technique to translate the original language writing, analyze it using context-sensitive grammar, and then generate the pronunciation using a computer [11]. This approach has the greatest accuracy in theory. The grammatical syntax of the natural language is, nevertheless, complex and difficult to condense. English translation strategies based on statistics have a comparatively low level of complexity. The statistical method of translation is more fluid than other approaches and has been the standard approach to machine interpretation for a long time [12]. Reinforcement Learning (RL) is the process of learning via errors of judgment, and its objective is to act in a way that will yield the greatest long-term gain, like the highest accuracy [13].

## Problem statement

Improper word choice, context-specific meanings, and ambiguity in word meaning are the main challenges in achieving high-quality English translations, which are focused on this research idea. Accuracy is a common problem with current translation methods because they can't handle semantic uncertainties and various linguistic contexts. This research presents a translation model that uses reinforcement learning and fuzzy semantic characteristics to improve the quality and accuracy of translations built on edge computing.

The suggested model addresses the increasing need for accurate and efficient English translations by enhancing the translation process with comprehensive bilingual corpora and sophisticated control techniques.

## Research questions (RQ)

RQ1: Can reinforcement learning policies be fine-tuned for domain-specific translation tasks? - Future research could focus on refining the optimal control techniques to customize translations in legal, medical, and other sectors, improving accuracy for specialized applications.

RQ2: How does the proposed FSLR-P2O model perform across multiple language pairs and different language families? - The future research scope could extend the model to languages with varying grammar structures, testing its adaptability and scalability beyond English-based translation.

RQ3: What is the impact of user feedback on enhancing translation quality? - Interactive learning approaches could be integrated to allow real-time adjustments to translation policies based on user preferences and feedback, increasing personalization of user experiences.

By addressing these research questions, future work can build on the current use of reinforcement learning and fuzzy semantics, further enhancing translation accuracy and adaptability.

## Main contribution

The main contribution of this study is as follows:

• Designing a corpus-based English translation model to enable efficient source and target language translation.

• Fuzzy semantic similarity is used to quantitatively evaluate the similarity or proximity of two words in a sentence pair by considering their semantic content.

• An optimal control policy is implemented through the RL-assisted P2O technique by an agent's state and action behaviour, and policy updates improve the translation accuracy and quality.

• Validate the FSRL-P2O scheme by translation accuracy and semantic similarity using fuzzy, BLEU, and NIST scores.

The rest of the article is outlined as follows: Review of the Literature: This section overviews earlier investigations and studies on corpus-based English translation and the application of fuzzy semantics. Proposed Design: describe how the proposed Fuzzy Semantic (FS) optimal control translation model was designed and put into training based on the P2O of

the RL algorithm, including how fuzzy decision processes were used. Results and Discussion: This part summarises the proposed system's deployment outcomes and contrasts its effectiveness with the existing English translation process. The results, as well as their consequences, are also thoroughly discussed. Conclusion and Future scope: The article's last section summarizes the fundamental discoveries and contributions made by the study and explores potential future lines of inquiry.

## 2. Related work

Enhancing translation accuracy, eliminating ambiguity, and raising the overall translation quality of the English translation model have all been the subject of related research in translation models and techniques. Numerous strategies have been investigated earlier, including statistical methods, models based on neural networks, and hybrid models that combine both approaches. This section discusses each existing research work, including its advantages and limitations where they are lacking. Hence, these limitations play a key role in the motivation of our research.

A deep neural network (DNN) is employed for training after processing and analyzing the input data; the neural network performs auto-tuning operations that aid in improving the model using the machine translation method [14]. The Song lyrics are input material in a novel corpus-based translation method for language 1 to language two translation. When the target language was created using the suggested model, the BLEU Level and word error rate improved by 40% more than the rule-based method.

Bi [15] applied a method for translating between English and other languages using Artificial Intelligence (AI) based on neural networks and intelligent knowledge bases. To find a more advantageous way to translate long English sentences based on the current English-Chinese machine translation. It is possible to enhance the tagging of parts of speech and rules to match more phrase patterns and enhance the accuracy of current machine translations. The achieved accuracy is 35.7%, which is not so high. The shortcoming is the N value suggested by Top-N may not have been properly chosen, or the k value in clustering may have been decided incorrectly.

Zhang and Liu [16] proposed a framework for translating English using NN with Fuzzy Semantic Optimal Control (NN-FSOC) to address the issue of inconsistent character traits in the answers. It works based on standard English translation knowledge for algorithm pretraining and uses optimal control and fuzzy semantic information for fine-tuning. An assessment network that can forecast future whole-sentence returns using deep inheritance features and a sequence-generating system that gives words probability distribution. The proposed system showed an increased accuracy of evaluation score of 95%, BLEU score of 65, and system generalization. As NN layers expand, computing complexity increases, affecting training processing time and speed.

Li [17] presented an Improved Generalized Maximum Likelihood Ratio detection (FSOC-IGLR) Fuzzy Semantic Optimal Control smart identification system for English translation. This technology marks thousands of English and Chinese phrases in a corpus to automatically search sentences. You can get incomplete speech recognition results. Voice recognition contributes. The updated algorithm achieves >95% recognition accuracy, resulting in 93%. Warning signs of translation consequences are incoherent. Second, the loose and compact arrangement of node points of control alternates emphasizes instability.

Yu and Ma [18] constructed a Deep Learning (DL) and intelligent recognition-based English translation model that estimated the phrase predecessor and subsequent probability of the Enhanced GLR (EGLR) technique using Quaternion Clustering (QC) and determined the English phrase corpus' part of speech. The best contextual characteristics are extracted using the feature extraction method. Merging the attention mechanism for English translation creates an artificial translation machine model. Excellent recognition accuracy, strong impact, and short duration improve English translation quality. Reference translation in training may cause tense inaccuracy.

Li [19] suggested using the English corpus for translation using FSO solution intelligent selection. The English corpus translation fuzzy semantic phrase attribute orientation framework exists. English translation fuzzy semantic intelligence

is the best solution variable computed using link association fuzzy semantic ontology attribute. English translation implements fuzzy semantic feature matching and adaptive target phrase registration. Results showed improved translation accuracy and feature match. Corpus-based methods can provide statistical data but may not capture the delicate contextual and cultural aspects needed for good translation.

Liu and Liang [20] created a successful workflow to obtain a parallel corpus in English and Chinese from the New England Journal of Medicine (NEJM) to train highly qualified translators and translations in the bio-medical platform. This corpus comprises about 100,000 sentence pairings and 3,000,000 tokens on both sides. The result showed improved BLEU translation quality by 33.1 for en to zh and 24.5 for zh to en directions. In addition, greater precision, low recall, error analysis of machine translation, and F1-score. The challenge showed that the biomedical domain is only partially generalizable by a baseline model (WMT18) trained on non-biomedical data.

Yuan et al. [21] offered a Fuzzy Algorithm-based English Translation system to partially eliminate Semantic Ambiguity (FAET-SA) in the translation process and analyze it from the corpus, grammar, syntactic base, and translation characteristics to fully comprehend its language characteristics. The Gaussian blur approach processes the features using image input and a recognition unit to enhance the translation's correctness. The performance results showed a higher translation accuracy of 87%, a recall rate of 90%, and quality of English translation without semantic ambiguity. The translation outcome is largely affected by image deformation and quality.

Jian et al. [22] added an attention mechanism to the traditional LSTM English translation model; the proposed model based on LSTM attention embedding improves source language contextual information representation, improving the English translation model's effectiveness and text quality. Scoring takes 4.923s, less than other models, and yields a higher BLEU score. Long words might increase alignment uncertainty and distract the attention mechanism, reducing translation quality.

Fan [23] suggested a sequential-to-sequential translation model based on a neural network model to improve translation accuracy by predicting language translation word order and creating a linear framework to implement this statement and allow semantic information extraction. Experimental results show that the preprocessing technique improves English-Chinese translation performance by 40 BLEU scores and reduces parameter scaling and model training time by 20%.

Zhang [24] Particle Swarm Optimization and a neural network-based English translation were discussed. This study integrates IoT, PSO, and Neural Network methods to evaluate students' English translation skills. The Neural Network algorithm employed in PSO uses particle coding, which this study describes. An example application assesses English translational teaching students. The PSO algorithm tracks student progress to accelerate English translation instruction. The researched findings create learning approaches and instructional materials for diverse learners. Second, this study establishes English-language translation practice. A PSO-enabled Neural Network can handle English language translation and instruction data by training in a global optimum state of PSO and minimizing training errors. This study compares the training and test samples' average mistakes using 5, 10, and 20 particles to assess the model's performance. The final results suggest that assessing students' translating ability is accurate. This study also analyzes the EFLT, ELLC, and EMPC data sets, which had detection accuracies of 0.84, 0.97, and 0.65, respectively.

Guo [25] discussed enhancing English machine translation with Deep Neural Networks. The language training model includes context recognition, and machine translation is altered slightly as the transfer learning goal is to increase translation performance. Job design includes word alignment optimization to boost transformer system efficiency. The suggested strategy reduces average alignment error rates by 8.1% in EnRo (English-Roman), 24.4% in EnGe, and 22.1% in EnFr compared to earlier methods.

Elakkiya et al. [26] released a new generation of Generative Advanced Networks (GANs) that use hyperparameter optimization to distinguish between hand and non-hand movements in sign language detection. The H-GANs operate in three stages: first, by modifying SVAE and PCA to decrease feature dimensions; second, by generating features using

Deep Long Short Term Memory (LSTM) and 3D Convolutional Neural Network (3D-CNN) as discriminators; and third, by optimizing and regularizing hyperparameters using Deep Reinforcement Learning. The system attains improved accuracy as well as recognition rates compared to cutting-edge classification techniques.

Natarajan et al. [27] established a new Neural Machine Translation (NMT) system that uses deep stacking Gated recurrent unit (GRU) algorithms to solve problems with translating unfamiliar words and terms not in the dictionary and deciphering word associations and linguistic structures in multiple languages. Sign language automates video creation with sign motions, outperforming previous methods. Testing the model with several sign language datasets improved translation results and lowered processing costs.

Rajalakshmi et al. [28] established a system that can classify the sign language spoken by people who are deaf or hard of hearing. Machine vision researchers face a formidable obstacle when deciphering distinct sign languages from static and moving images. A Hybrid Neural Network Architecture is suggested to address these challenges in recognizing Isolated Russian and Indian Sign Language. For static gesture recognition, the framework uses 3D Convolution Net. It employs semantic spatial multi-cue feature detection and extraction for dynamic gesture recognition. The proposed study also develops a new dataset for Russian and Indian Sign Language that includes multi-signer, single-handed, and double-handed isolated signs.

In the present research, edge computing improves machine translation accuracy and efficiency. This performance comparison shown in Table 1 shows edge computing and cloud-based performance indicators, including latency, throughput, and resource utilization are shown in this table. Edge computing is faster and more efficient in real-time, while cloud-based systems are better at bulk data processing and resource management. These comparisons are corroborated by [16,18,23], and [25].

It comes to an understanding that while machine translation has made many strides for corpus-based English translation, there is still much space for improvement. The existing algorithms are discussed with their limitation where the proposed system comes into the picture to enhance the challenge metrics like accuracy, quality, and semantic update.

The suggested translation model based on edge computing is a significant development in translation technology. It enhances accuracy by incorporating Fuzzy Semantic characteristics for a more nuanced comprehension of semantic material. It also integrates Proximal Policy Optimization as well as Reinforcement Learning approaches for flexibility and optimization. The model's performance is fully revealed through metrics such as NIST scores, BLEU, and fuzzy semantic similarity in its evaluation.

## 3. Proposed scheme

Corpus-based English translation systems are distinct from rule-based or dictionary-based translation procedures. Using statistical and ML techniques to find similarities and then arrive at translation judgments per actual language practice. Semantic recognition and evaluation of features are essential in translating English using computer software. An English translation model is generated after extracting semantic information from the English context and corpus. The sources of English translation difficulties with accessibility are typically identified. The fuzzy semantic optimal control technique uses RL-assisted P2O while employing English corpus to accomplish translation work. The subjective nature of the person who wrote the original text is reflected in the logical organization as well as content development of the English translation,

**Table 1. Performance Comparison Between Edge Computing and Traditional Cloud-Based Approaches.**

| Ref. Nos | Performance Metric | Edge Computing | Traditional Cloud-Based Approaches |
|---|---|---|---|
| [16], [18,23] | Latency | Lower latency due to proximity to data | Higher latency due to longer network paths |
| [16], [23] | Throughput | Higher for localized data processing | Higher for bulk, non-real-time tasks |
| [18], [23,25] | Resource Utilization | Efficient for real-time, decentralized processing | Centralized resources for large-scale batch processing |

which raises the standard of automation and intelligence of English translation. Enhancement of translation accuracy by leveraging large bilingual corpora, considering fuzzy semantic (FS) properties, and optimizing translation output via Reinforcement Learning and Proximal Policy Optimisation (PPO) technique. The first step is to collect the corpus data and perform any required preprocessing and feature extraction. Fuzzy semantic similarity takes the preprocessed sentences as input and attempts to remove uncertainty by calculating the Jaccard similarity coefficient to determine the degree of semantic similarity between any two linguistic fragments (words, phrases, or whole sentences) in a translation.

Parallel corpora must be synchronized at the sentence or phrase point to create a translation system. Finding matching sequences in the source and target languages is the alignment process. Following alignment, the corpus is utilized to retrieve translation units that help in translation, including word combinations, phrases, and possibly lengthy segments. Character and word embeddings are two examples of implicit linguistic properties that can be acquired universally from a huge external corpus. A type of machine learning called reinforcement learning entails instructing an agent to arrive at choices successively based on input from its immediate surroundings. The translation model can be considered an entity in the corpus-based English translation, and the translation procedure can be conceptualized as a consecutive choice contention. Fig 1 illustrates the proposed scheme translation model for corpus-based English.

The proposed edge computing-based translation model uses Fuzzy Semantic (FS) properties, Reinforcement Learning (RL), and Proximal Policy Optimization (PPO) techniques to address uncertainties and ambiguities in natural language translation. FS properties quantify semantic similarity between words, phrases, or sentences to mitigate ambiguities. RL handles out-of-domain data and low-quality references in translation, allowing the model to learn and adapt through trial-and-error interactions. PPO optimizes the translation policy efficiently and stably, ensuring policy updates do not deviate significantly and maintaining quality and coherence. This comprehensive approach combines fuzzy logic, machine learning, and optimal control theory to improve accuracy and semantic coherence in machine translation

## 3.1 Corpus acquisition

Collect a significant linguistic corpus of comparable synchronized texts works in the source language ($S$) and their target English translations ($T$). The corpus must encompass a range of genres and subjects to guarantee full representation. A

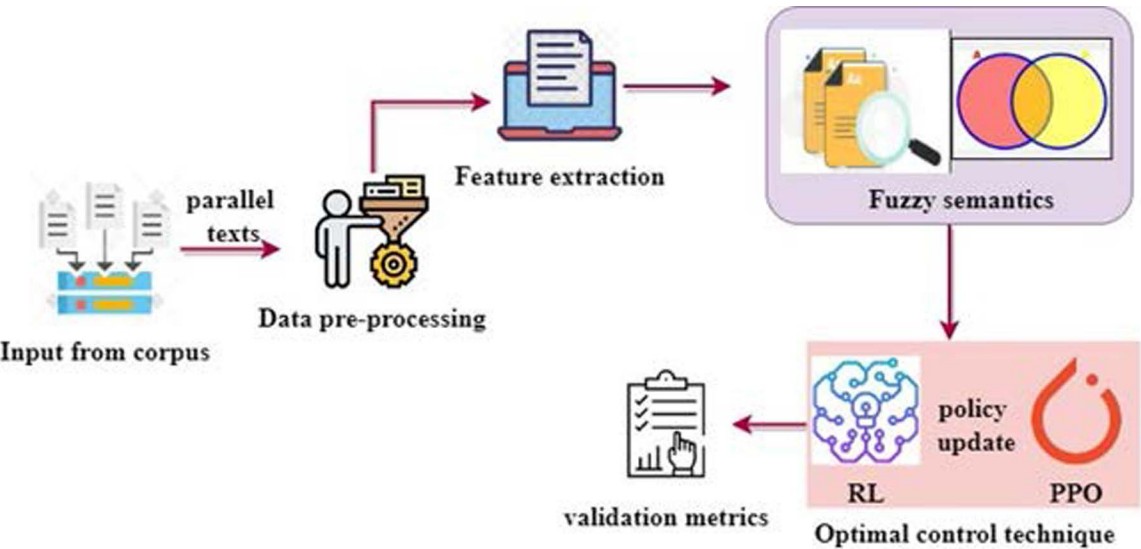

**Fig 1. Proposed scheme.**

corpus assemblages words or documents in one or more languages as a linguistic resource for study and translation. The source texts are in the original language, English, and their equivalent translations into Japanese make up the parallel corpus.

## 3.2 Data preprocessing

The data are taken from the machine translation Kaggle dataset [27]. Every data and task needs different preprocessing. Preprocessing includes syntax and sentence classification analysis before extraction. In this method, the words or features of a sentence, document, website, etc., are extracted and then categorized according to the frequency of use. As shown in Fig 2, an initial step is to preprocess the data gathered from the corpus to remove unwanted special characters, punctuation marks, and tags to separate the sentence into tokens. Tokenization, phrase segmentation, and linguistic preprocessing are just a few preprocessing techniques used to clean and prepare the acquired corpus data. Tokenization is the initial procedure of breaking up a long text string into tokens. Tokenization is crucial for preparing the text data in the framework of corpus-based English translation to implement translation techniques or carry out any further analysis.

Original text in the source language: we're moving to native!

Whitespace: tokens ["we're," "moving," "to," "native!"]

Punctuation: tokens ["we," "'re," "moving," "to," "native," "!"]

The punctuation marks in the target language have many forms like "、 。【】「」『』…・ ヽ () 〜?!.:,;.". Unlike the English language, Japanese punctuations are treated using janome_tokenizer ().

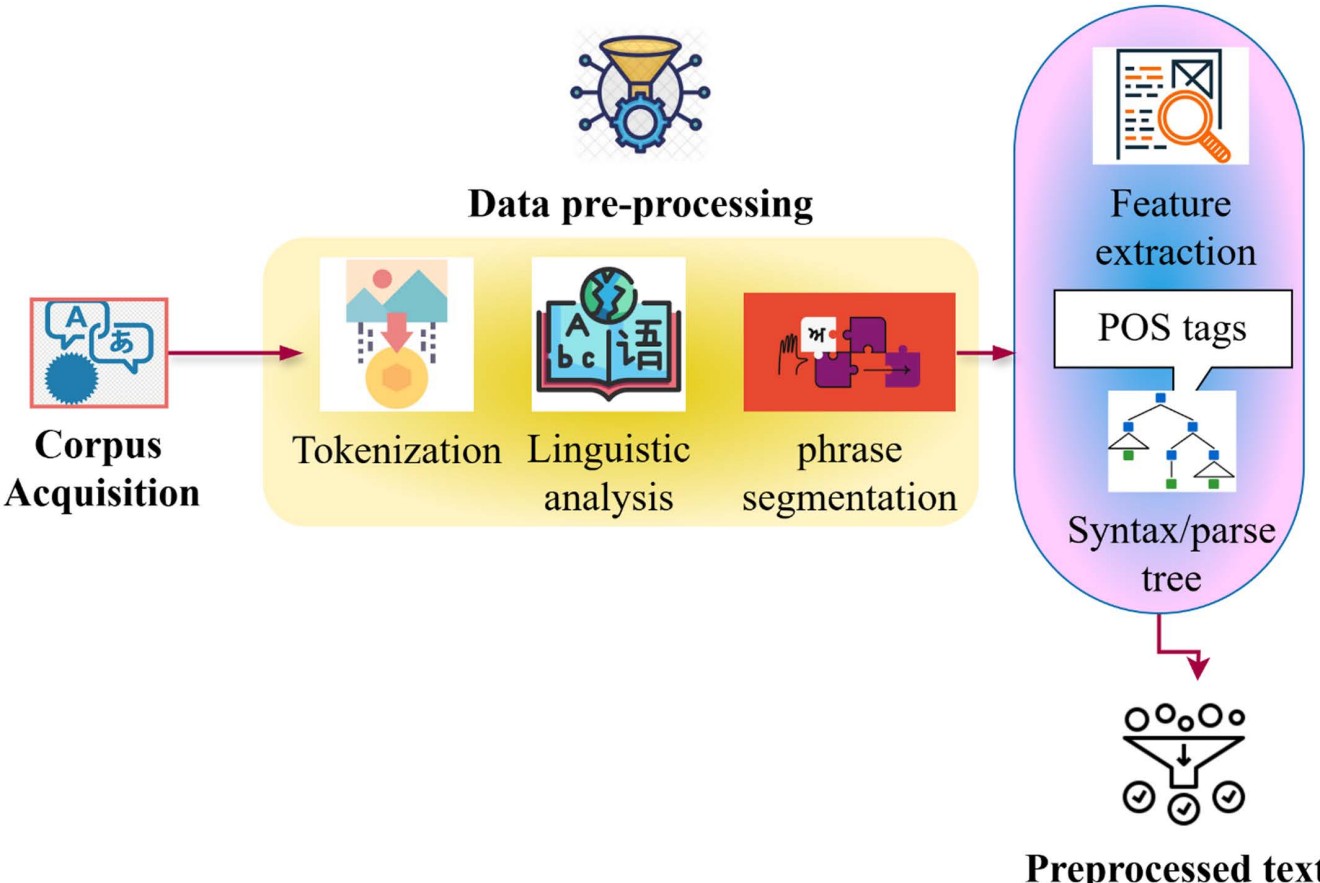

**Fig 2. Preprocessed texts.**

## 3.3 Linguistic analysis

The rate of occurrence and arrangement of phrases and words across all languages are better captured by language analysis.

Fuzzy rules for analyzing linguistic variables: Instead of crisp sets, words and phrases have been rendered as fuzzy sets with progressive merging structures of categories of language captured by fuzzy sets. Each word or idea has a membership function that specifies how much it belongs to specific groups or classes. The following example indicates the fuzzy rules for determining the verb, noun, and adverb linguistics classes. The rules analyze particular circumstances based on word or sentence features and part-of-speech characteristics using logical expressions like IF, AND, and NOT. The translation model may determine the linguistic classification of a given sentence by using these fuzzily defined principles.

Fuzzy Rule1-verb

IF (word ends with "-ing" OR "-ed") AND (NOT noun OR NOT adjective) THEN verb.

Fuzzy Rule 2 – Noun

IF (word ends with "-tion" OR "-ment" OR "-city") THEN noun.

Fuzzy Rule 3 – Adverb

IF (word ends with "-ly") THEN adverb.

IF (word accompanies with a verb) AND (NOT identified as part-of-speech) THEN adverb.

Based on the given fuzzy rules, fuzzy sets described their membership values and variable linguistic terms. In general, the membership function described for verbs with a linguistic variable high is given by Equation (1).

$$\mu_{high}(verb) = max\left(0, \left(1, \left(\frac{(x-1)}{0.5}, \frac{(1-x)}{0.5}\right)\right)\right)$$

(1)

Since the fuzzy set contains verb as {0,0,0.5,1,1,1,0.5,0,0} with very low as [0,0,0.5], low as [0,0.5,1], fair as [0.5,1,1], high as [1,1,1] and very high as [1,1,0.5], the corpus-based English translation model can manage linguistic ambiguity and arrive at fuzzy conclusions about a word's linguistic category by using these fuzzy sets. The translation model may assess the extent that indicates a phrase belongs to a specific linguistic class depending on the supplied input value $x$ by applying these triangular fuzzy membership coefficients. The membership values are determined by the equations using mathematical procedures, including min, max, and calculation by arithmetic.

The sentence alignment of the translation is arranged clearly. Use statistical analysis to find similarities and relationships between English and Japanese in the aligned corpus.

## 3.4 Feature extraction

Linguistic information can be retrieved through parallel statements to improve the translation model. Part-of-speech (POS) labels, syntactic parsing trees, named entities, and other pertinent linguistic data are a few of these features. Such traits improve translation by adding context. In a sentence, POS tags indicate a word's grammar. These labels show the grammatical structure and semantic relationships. The sample contains nouns (N) identifying people, places, and objects with determiners. Verb(V) is action, event, state. The adjective modifies the noun. ADV describes verbs, adjectives, and adverbs. To eliminate noun repetition, use pronouns. Each word in a phrase can be tagged with a POS label to recover such tags. They illuminate text grammar and semantics, enabling more advanced analysis and predictions. Syntax trees, also known as parse trees, map the term "Corpus-based English translation," as seen in Fig 3. They show a sentence's root node and its words and phrases' grammatical functions and dependencies. Translation is easier using syntactic trees, which group similar words and sentences from both languages. Corpus and translation are nouns. English is sometimes used as an adjective or noun. The adjective based comes from the word base. More exact and accurate translations are

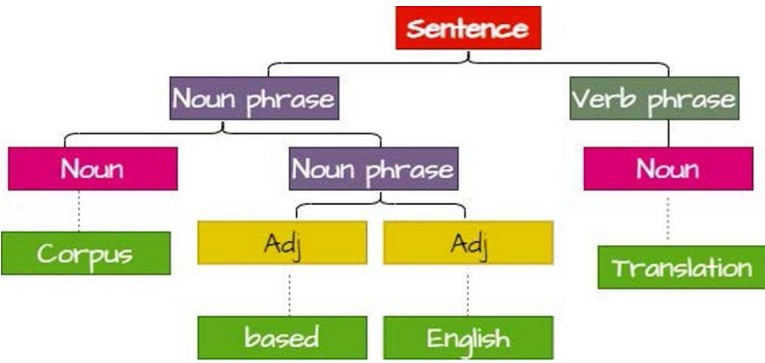

**Fig 3. Syntactic tree structure.**

possible using syntactic trees, which keep vocabulary-grammatical relationships. Syntactic trees ensure structural coherence by ensuring translated information follows the source's semantic structures.

Fig 3 shows the syntax tree representation of "Corpus-based English translation," also known as parse trees. They show how words and phrases in a sentence form a root node and interact based on their grammatical roles and dependencies. Syntactic trees group similar words and phrases in source and target languages to simplify translation. The corpus and translation are nouns. Depending on usage, English is an adjective or noun. Based is an adjective since it comes from base. Syntactic trees preserve vocabulary-grammatical relationships during translation, resulting in more accurate translations. Syntactic trees can maintain structural coherence during translation by ensuring that the translated material matches the source semantically.

### 3.5 Fuzzy semantic similarity analysis with ranking method

Fuzzy semantic similarity metrics compare phrases, words, and sentences. These evaluations account for linguistic categories' fuzziness and differences in membership. Use fuzzy and semantic interpretation to accurately reflect the original text's contextual data and linguistic changes. Long-range dependencies and contextual interactions between words in this model are captured well by the sequence-to-sequence mechanism during translation. As an input layer, fuzzy semantic similarity measurements feed the attention mechanism on word or phrase semantic importance. This layer computes fuzzy semantic similarity and semantic overlap between input sequences and training corpus. The translation model uses this attention mechanism to weigh the value of words in the sentence. Handling translation ambiguities requires this method. PPO optimizes translation outputs using predetermined incentive functions, including fuzzy semantic similarity, BLEU scores, and other performance indicators.

Evaluation and comparison of fuzzy semantic qualities are involved in fuzzy semantic decisions. As an illustration, if employing a scoring-based strategy, a simple Equation (2) for grading can be:

$$\sigma_s = \alpha J(S, T) + \beta B + \gamma C \tag{2}$$

The weighting parameters α, β and γ denote the relative weights or order of each fuzzy semantic characteristic in the final score. The weights might be changed based on the particular specifications and choices of the translation process.

$J(S, T)$ denotes Jaccord's measure of similarity. It compares the similarity between groups of phrases or n-grams instead of the full corpus. Determine how similar each group of phrases or n-grams is among the source language ($S$) as well as the language used for translation ($T$) using the Jaccard method. The dimension of the sets' intersection $S \cap T$ denotes the commonly shared words divided by the total dimension of the pair's union. $tot_{S,T}$ To determine the Jaccard

resemblance level. Create a threshold to specify the degree of similarity necessary for a given translation to be accurate. Although the Jaccard similarity coefficient approach is efficient and widely used, the similarity findings are very profound regarding the total number of words in the text, reducing the translational uncertainty. Translations that score higher on the Jaccard similarity scale are considered more similar and highly acceptable and are calculated using Equation (3).

$$J(S, T) = \frac{|S \cap T|}{|tot_{S,T}|}$$

(3)

As shown in Equation (3), where $B$ denotes fluency as a measure of the translation's linguistic accuracy or grammar, vocabulary, and idioms. It can be assessed using both general criteria and language-specific standards.

The translation's coherence is represented as $C$ defines the consistent logical flow of words in a sentence. It can be evaluated using the original texts or semantic coherence. This element evaluates how smoothly and naturally the target text reads to native speakers.

Ranking: Sort the translations in order of decreasing score ($desc$) based on their scores in response to the similarity score identified by Jaccord. Equation (4) represents the top position awarded to the translation with the biggest score, and the remaining translations are sorted in order of score.

$$R(S_i, T_i) = sort(desc)J(S, T)$$

(4)

As inferred from Equation (4), the chosen biggest score value of the concerned translation is considered the suitable translation model. These scores indicate the quality or acceptability of each translation.

The fuzzy semantic resemblance component provides the measure of semantic similarity between sentences in the source and target languages. It can estimate overlap by determining common characters and the longest matching sequence. Tokenization, the LCS method, the computation of the Jaccard similarity coefficient, and parameter changes are all part of the training process. This enables the component to generalize to sentence pairings not seen during inference by teaching it the mapping between sentence pairs and their matching scores.

**Calculation of the Jaccard Similarity Coefficient:** The Jaccard Similarity Coefficient is a method for calculating sentence similarity involving tokenization techniques, n-gram representations, weighting schemes, preprocessing steps, and a similarity threshold. Its performance is influenced by the choice of similarity threshold, which should be adjusted based on the translation task's specific requirements.

**Example:** Word-level Jaccard Similarity Consider the following source sentence (S) and its target translation (T): S: "The quick brown fox jumps over the lazy dog." T: "La rapide renarde brune saute par-dessus le chien paresseux."

Tokenizing the sentences into words, get: S = {"The", "quick", "brown", "fox", "jumps", "over", "the", "lazy", "dog"}. T = {"La", "rapide", "renarde", "brune", "saute", "par-dessus", "le", "chien", "paresseux"}

The intersection of the two sets (shared words) is: $S \cap T$ = {"quick", "brown", "jumps", "over", "the", "lazy", "dog"}. The union of the two sets (total unique words) is: $tot_{S,T}$ = {"The", "quick", "brown", "fox", "jumps", "over", "the", "lazy", "dog", "La", "rapide", "renarde", "brune", "saute", "par-dessus", "le", "chien", "paresseux"}

Applying the Jaccard similarity formula: $J(S, T) = \frac{|S \cap T|}{|tot_{S,T}|}$ = 7/ 18 ≈ 0.389

These examples demonstrate that tokenizing source and target sentences calculate the Jaccard Similarity Coefficient at different granularities (words or characters). The fuzzy semantic resemblance component of the translation model can use the similarity scores to quantify phrase overlap or resemblance.

### 3.6 Optimal control technique using RL-assisted P2O

Use optimal control approaches to streamline the translation procedure while taking the quality of translations, proficiency or fluency level, and consistency into account. Reinforcement learning and the PPO algorithm can be combined to optimize the linguistic

policies for improving translation accuracy based on rewards and input received during the translation procedure. In a manner related to how optimal control techniques iteratively alter control inputs to maximize the overall behaviour of the system, PPO functions using trial-and-error relationships with the corpus environment. The main contribution of this paper is using reinforcement learning to improve translation for language transferring. RL depends on monolingual instead of bilingual spoken language understanding corpora. This study trains the policy network using the decentralized and distributed proximal policy optimization, which consists of an RL network loss, policy network loss, and an entropy loss to encourage exploration of machine translation.

Step 1: Create a reinforcement learning-based translation model, as shown in Fig 4, to perform translation among linguistic choices and act as an agent. The agent aims to discover the best strategy for optimizing translation quality or other established parameters. Initialize the reward structure based on fuzzy semantic accuracy, BLEU score, and fluency to calculate the translation quality output. Next, update the current state of the translation procedure as a state space that involves $S$, the history of translation, and words or phrases in the sentence. Then, the action space relevant to an agent's translation task consists of choosing a single word or sentence from a predetermined vocabulary, using particular fuzzy translation guidelines based on semantic analysis, using fuzzy rules for handling linguistic variable patterns or making any other significant translation choice.

Step 2: Employ the P2O technique for training the translation policy.

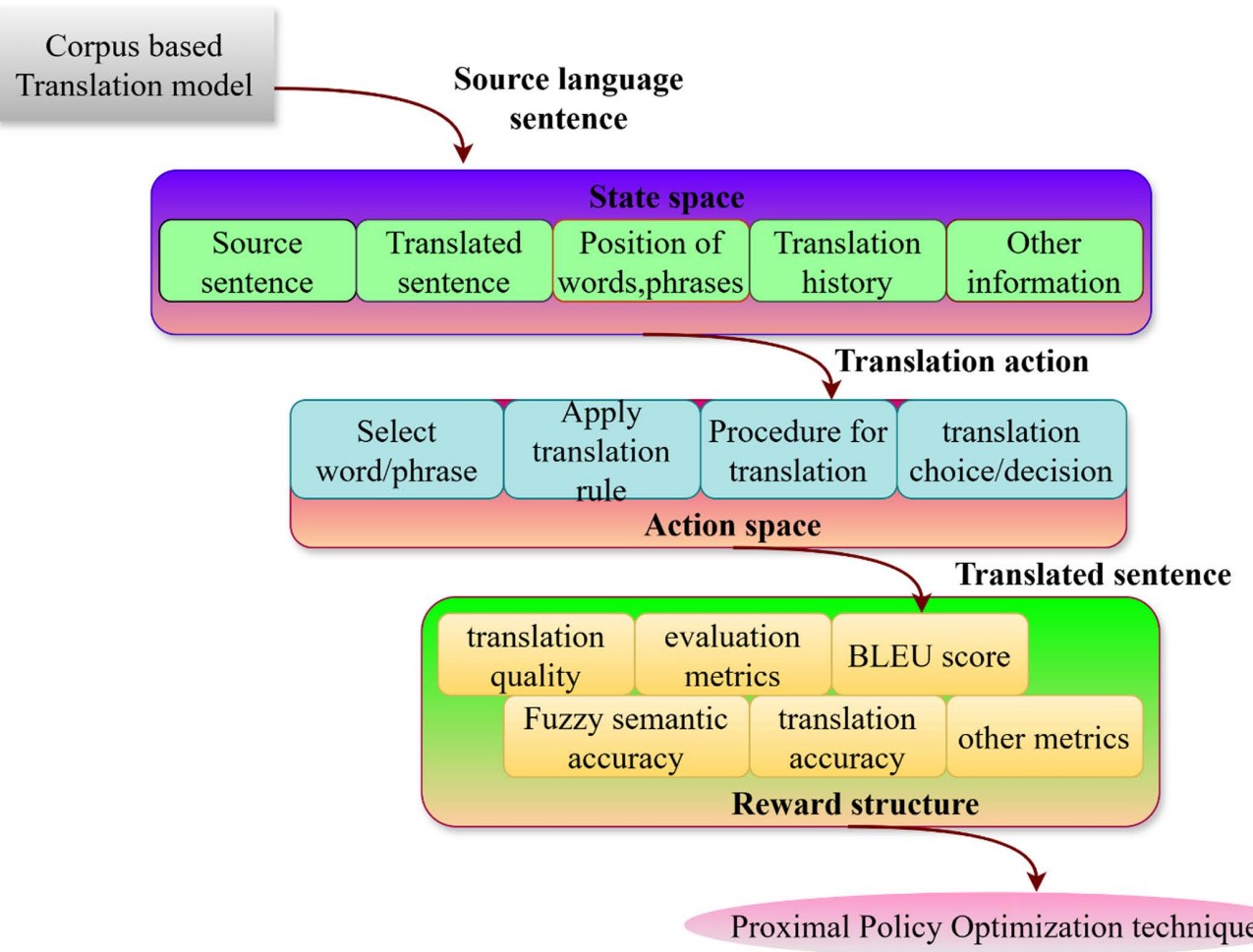

**Fig 4. Reinforcement learning environment for English translation.**

P2O works well for continuous control challenges, making it appropriate for choosing phrases, sentences, or translation procedures as part of the translation policy optimization process. Utilize a constant training procedure where the agent updates its translation policy, gains understanding through the translation instances, and gradually improves its translation decisions. The advantage is the disparity between a translated sentence as a reward, such as the BLEU score, and an existing initial source reward value for a created translation. The benefit aids in evaluating the translation's quality compared to the benchmark and directing improvements to the translation model. The model can grasp the effects of its activities and modify its translation method by estimating. $adv_n$ Using Equation (5).

$$adv_n = reward(BLEU\ score) - baseline\ value \tag{5}$$

Step 3: Setting up policy parameters and evaluation:

Equation (6) defines the policy gradient ($pg$) as the process of adjusting the policy so that good or correctly translated sentence actions result in maximizing the accuracy are sampled more often in the next possible sentence pair to manage the likelihood distribution of activities and maximize the anticipated return while enhancing translation quality.

$$pg = \nabla_\theta [\log \log\ (\pi\,(state_n))\ * adv_n] \tag{6}$$

As discussed in Equation (6), where $\nabla_\theta$ denotes the gradient changes in response to the parameters of the translation model. $\pi(state)$ Represents the likelihood of choosing an action in response to the corresponding state for $n$ number of sentence translations with the fuzzy semantic policies. $adv_n$ Represents the advantage if $adv > 0$; this is the best translation outcome for a given sentence pair compared to other possibilities in the current translation state.

Step 4: Policy improvement

A Clipped surrogate objective function from Equation (7) acts as a new objective function used by PPO to maintain consistency and avoid significant policy modifications of translation behaviour by use of a clip. The surrogate goal is maximized using the PPO algorithm to modify the policy settings.

$$obj_{clip}(\theta) = min(rt(\theta) * adv_n, clip\,(rt(\theta), 1 - \epsilon, 1 + \epsilon)\,adv_n) \tag{7}$$

As found in Equation (7), where $rt(\theta) = \frac{\pi(state_n)}{\pi_{old}(state_n)}$ The likelihood of the translation acting as a state in the current corpus model is divided by the previous translation behaviour identified for the parallel text. The hyperparameters of the clip thresholds are given by $\epsilon$.

Step 5: Training and iteration update

By adding feedback from the optimal control environment, such as rewards, and encouraging better translation choices, the update stage seeks to optimize the translation model's attributes. Employing the calculated $pg$ or the $obj_{clip}(\theta)$, the translation model's attributes get updated using Equation (8).

$$\theta_{new} = \theta_{old} + \lambda * [pg|obj_{clip}(\theta)] \tag{8}$$

As shown in Equation (8), the old and newly updated attributes of the corpus-based English translation model are given as $\theta_{new}$ and $\theta_{old}$. The learning rate is modelled as $\lambda$ for the model. The translation model can reach an ideal policy that maximizes translation quality or other predetermined criteria through an iterative approach. With the help of fuzzy semantic properties, the model analyzes surrounding words to determine their meaning. For instance, the model analyzes surrounding words to determine their meaning. It evaluates the relevance of words based on context, distinguishing between meanings. The model is trained on diverse examples using reinforcement learning through learning mistakes through feedback. It adjusts its translation strategies based on rewards and penalties. The model refines its translation policies

over time using PPO, improving its ability to differentiate meanings based on context. The translation example is given below:

The Akash is translated to "空は晴れています。"

Akash is playing soccer is translated to "アカシュはサッカーをしています。"

The model can resolve ambiguities by learning from its uncertainties and better grasping the context.

---

```
Pseudocode 1: FSRL-P2O
Input: Bilingual corpus (source S, target T)
Output: Optimized translation model
  Step 1:// Preprocess corpus data
    initialize corpus_data = load_data(S, T)
    perform tokenization using janome_tokenizer(corpus_data)
    calculate linguistic analysis using Equation (1)
    extract features using POS_tagging(corpus_data
  Step 2:// Calculate Fuzzy Semantic Score
```
$\sigma_s = \alpha J(S, T) + \beta B + \gamma C$ using Equation (3)
```
  Step 3:// Fuzzy Semantic Similarity Analysis
    for each sentence pair (S, T) do
```
calculate Jaccard Similarity $J(S, T) = \frac{|S \cap T|}{|tot_{S,T}|}$ using Equation (3)

Rank translations $R(S_i, T_i) = sort(desc)J(S, T)$ using Equation (4)
```
    end for
  Step 4://Initialize RL environment
    initialize state_space = [translation_history, current_word]
    initialize action_space = [word_choices, translation_rules]
    initialize hyperparameters: learning_rate = 0.01, batch_size = 32, dropout_rate = 0.3
    set boundaries = {learning_rate: [0.001, 0.05], dropout_rate: [0.1, 0.5],
    batch_size: [16, 64]}
  Step 5:// PPO Training Loop
    for each iteration, do
    for each batch of sentence pairs, do
    translate sentences using current policy
```
calculate $adv_n = reward(BLEU\ score) - baseline\ value$ using Equation (5)

calculate $pg = \nabla_\theta[\log \log (\pi(state_n)) * adv_n]$ using Equation (6)

calculate $obj_{clip}(\theta)$ using Equation (7)

perform translation probability $rt(\theta) = \frac{\pi(state_n)}{\pi_{old}(state_n)}$

Update policy parameters $\theta_{new}$ using Equation (8)
```
  Step 6:// Evaluate model
    calculate BLEU and NIST scores
    assess fuzzy semantic similarity
  Step 7: if not converged, then
    go to step 4
  else
    return optimized model
```

---

The FSRL-P2O machine translation pseudocode 1 employs fuzzy semantic analysis and reinforcement learning optimization. Tokenization, language analysis, and POS tagging are among its seven steps for bilingual corpus data. The algorithm quantifies source-target language element semantic relationships using the Fuzzy Semantic Score. Proximal Policy Optimization (PPO) training loops ensure core learning. Complex semantic nuanced translation challenges benefit from this hybrid method.

Step 6: Evaluation

Use the evaluation results to improve the translation model. Change hyperparameters like learning rate, batches, and periodicity to enhance the model's translation accuracy and quality. Create an independent validation or test set of S sentences and T translations. Assess the trained model's translation accuracy and quality. Maintaining high

translation quality and adapting to changing language trends and needs requires continual tracking, evaluation, and model improvement. Iteratively train and enhance the translation model using the corpus and RL methodologies like P2O. As a result, the model may learn and improve translation choices, producing more accurate and high-quality output.

Table 2 shows how hyperparameter adjustment optimized the model's performance for accurate predictions in this research. A learning rate of 0.01 balanced speedy convergence with model stability. The 250 epochs allowed the model to understand data patterns without overfitting. A 32-batch size improved computing efficiency and training stability. Avoiding overfitting with a dropout rate of 0.3 improved the model's generalization to fresh data. With a hidden layer size of 128, the model could properly capture complex patterns, improving its prediction accuracy. While corpus-based translation aims to create dependable translations, human translators often must polish and improve the outcome. Quality may be improved by post-processing methods such phrase reordering, word selection, and styling tweaks. Manual assessments or computational indications using Bilingual Evaluation Understudy measure translation integrity, fluency, and readability.

Language ambiguity, contextual errors, and named entity identification problems are common translation model errors. It mistranslates "Akash" and colloquial phrases. Using too many n-grams, insufficient contextual knowledge, and insufficient training data are common culprits. Higher-quality training datasets, contextual awareness models, fluency-optimized reinforcement learning, culturally relevant examples, and named entity identification systems can reduce these errors. These changes will increase the model's translation accuracy and efficiency.

Different industries like media, e-commerce, education, healthcare, and tourism could benefit from using the FSLR-P2O translation algorithm. This allows for precise translations of product descriptions and reviews, medical system records, and e-learning platforms with multilingual support for providing real-time global learners and consumer interactions. The approach provides more dependable and speedy multilingual support for enhanced user interactions. As a result, accessibility is improved across industries, user experiences are improved, and global communication is fostered across these industries.

The proposed scheme performs a corpus-based English translation model using four phases. Initially, the sentences of parallel texts are collected from a corpus of English and Japanese as a source and target. The data preprocessing and feature extraction include several processes to clarify the sentence for performing the translation. Then, the processed sentences utilized a fuzzy algorithm to eliminate semantic uncertainties, clear understanding of vocabulary, and meaningful translation of phrases and verbs using semantic similarity with the Jaccard index and fix the rank based on the index value to find out the closeness of similarity between the machine translated and reference translated sentence. The optimal control techniques are implemented to identify the accuracy and quality of the translation by frequently updating the policy of P2O in RL. Then, the proposed scheme is evaluated using various metrics to prove its effectiveness, which will be discussed in the next section.

The suggested translation model uses optimum control approaches such as proximal policy optimization (PPO) and reinforcement learning (RL) to address machine translation issues. While PPO guarantees stable policy updates, RL enables the investigation and modification of translation strategies. Even without ideal reference material, this method generates high-quality translations, adjusts to new language distributions, and learns from errors. Adding domain-specific metrics or quality scores to the reward function can enhance performance.

The Edge Computing-based English Translation Model (FSRL-P2O) addresses idioms, sarcasm, and domain-specific terminologies using the following mechanisms:

**Table 2. Hyperparameter Tuning Summary and its Impact on Model Performance.**

| Hyperparameter | Optimal Value | Impact on Performance |
|---|---|---|
| Learning Rate $\lambda$ | 0.01 | It is possible to achieve stable and fast convergence. |
| Number of Epochs/iterations | 250 | Provide effective learning without overfitting problems. |
| Batch Size | 32 | Improved training performance stability and validation |
| Dropout Rate | 0.3 | Reduced overfitting and enhanced generalization on unseen data. |
| Hidden Layer Size | 128 | Captured complex data patterns |

- Idioms: The fuzzy semantic similarity analysis captures contextual and semantic overlaps, allowing the model to identify idiomatic utterances. It quantifies idiom semantic links using the Jaccard similarity coefficient, taking linguistic variances into account to retain meaning throughout translation.

- Sarcasm: The Sarcasm model uses Reinforcement Learning (RL) and Proximal Policy Optimization (PPO) to acquire contextual nuances over time. Sarcasm recognition could be improved with specialized datasets, but it refines its translation policies by examining sentence structures and semantic contexts during training rounds.

- Domain-Specific Terminologies: – The system integrates domain-aligned corpora and fuzzy rules for specialized terminology. Language elements are aligned by semantics, and translation policies are adjusted using RL to ensure consistency in technical or specialized contexts.

## 4. Results and discussion

### 4.1 Data study

The employed data source consists of 55463 bilingual sentence pairs taken from English to Japanese language corpus from [29]. The source language is English, and the target language is Japanese; initially, it must handle misspelt words and remove punctuation. Training models that translate sequences from one domain (for example, English sentences) to sequences in another domain (for example, Japanese translations of those English phrases) is the goal of sequence-to-sequence learning (Seq2Seq). The goal is to convert sample languages, such as Japanese sentences, into English with training and testing datasets. The training data contains 51450 images, and the testing data contains 2144 images. The sentence pair is categorized into train_test_split. The corpus's inclusion of modern and historical literature provides a diverse linguistic and cultural context for Natural Language Processing (NLP). The total vocabulary size used for the basic language is 9646, and the target language is 14403. The sample translation from the source to the target language taken from [30] is given below:

Input: English Sentence
here I come
Output: Japanese Translation
ここ に 来 た の
Japanese Reference Translation
いま 行き ます

Tables 3 and 4 show the vocabulary statistics and mean and standard deviation for translation length tasks.

**Table 3. Vocabulary Statistics.**

| Language | Vocabulary Size | Mean Word Length | Standard Deviation (Word Length) |
|---|---|---|---|
| English | 9,646 | 4.7 characters | 1.8 characters |
| Japanese | 14,403 | 2.5 characters | 1.2 characters |

**Table 4. Mean and Standard Deviation for Translation Lengths.**

| Dataset | Mean Sentence Length (English) | Mean Sentence Length (Japanese) | Std Dev (English) | Std Dev (Japanese) |
|---|---|---|---|---|
| Training | 12.3 words | 8.7 words | 3.4 words | 2.8 words |
| Testing | 11.8 words | 8.5 words | 3.2 words | 2.7 words |

## 4.2 A comparative study with existing algorithms

To emphasize the proposed model's advantages and enhancements, contrast its performance with those of other cutting-edge machine translation models. The existing algorithms, such as NN-FSOC [16], FSOC-IGLR [17], and FAET-SA [21], are taken for comparison study based on metrics like translation accuracy, BLEU score, NIST score, and fuzzy semantic similarity. The evaluation parameters differ because NIST and BLEU scores focus on improving translation quality. The NIST provides informativeness, so the number of sentences helps assess how well the model conveys essential information. BLEU measures n-gram precision, which is influenced by sentence length because longer sentences provide more contextual information and help to improve translation accuracy. Thus, each metric evaluates distinct qualities of the translation, which is why different parameters are used.

  (i)  **Translation accuracy analysis.** Fig 5(a) demonstrates that the proposed FSRL-P2O translation accuracy is comparably higher than the existing models since it comes up with the optimal control policy update based on the state and corresponding action performed by the agent during the training period analyzed. The training parameters are frequently updated and hyper-tuned based on the objective function to meet the quality of the translation accuracy. The highest accuracy is obtained at 97.4%, whereas FAET-SA performs less.

  Fig 5(b) compares various methods for achieving accuracy across different iterations. SDI-RSIR-HNN has the lowest accuracy, followed by GAN-CSLC, S2SG-HNMT, and FSRL-PSO. As iterations increase from 50 to 250, accuracy generally improves, except for a slight dip in FSRL-PSO at 150 iterations. The FSRL-PSO method, likely a proposed edge computing-based translation model, outperforms the other methods, especially at higher iteration counts.

  (ii)  **Bi-Lingual Evaluation Understudy (BLEU) Score.** BLEU makes automatic evaluation of machine-translated text possible (Bi-Lingual Evaluation Understudy). A machine translation's BLEU score is between 0 and 1, indicating how close it is to matching a reference translation database [28–29]. The BLEU score is essential for assessing sentence translation accuracy compared to human-generated reference translations. Shorter interpretations do not translate well, whereas longer interpretations yield reliable results. Standard BLEU Score readings range from 0 to 1, and their percentage can be computed by multiplying that number by 100. It has been found that the model is more accurate when increasing the bleu score value, and if the score decreases, the translation model is less accurate. Equation 9 defines the BLEU Score formula as follows:

$$BLEU_{score} = min\left(1, \frac{O_l}{R_l}\right)\left(exp\left(sum(wt_i * p_i)\right)\right)$$

(9)

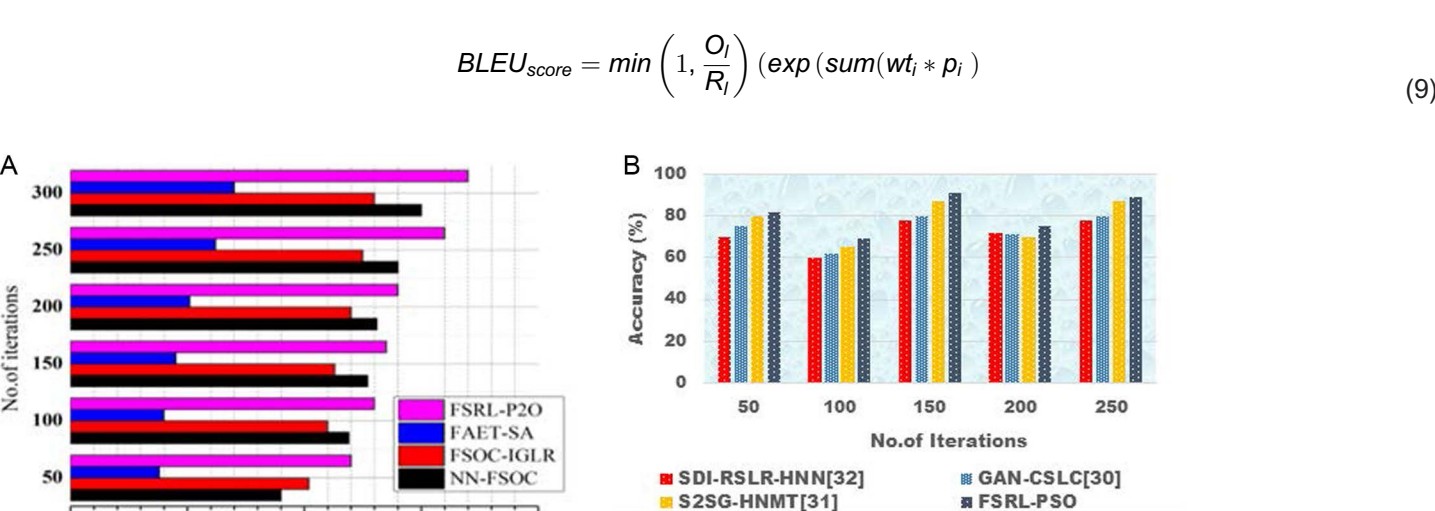

**Fig 5.  (a). Comparison of Translation Accuracy.** (b). Additional Algorithms for Comparison of Translation Accuracy.

Where $p_i$ represents the precision for N-grams with order $i$ and weight $wt_i = \frac{1}{N}$ presents in the ratio between reference and machine translation.

Fig 6 shows the experimental results that the suggested FSRL-P2O model has a higher BLEU score than the existing models, showing that it is more efficient at translating lengthy words and has significantly enhanced model performance. Hence, the best BLEU score for training is obtained using an optimal translation strategy and necessary optimization techniques. After a training of the training set text, the translated sentence is taken according to the test results to verify that the modified fuzzy membership degree value of the keywords is updated with the necessary. $wt_i = \frac{1}{N}$ translation can be performed for the corresponding linguistic variables. It has been discovered that as the BLEU score value rises, the suggested translation model becomes more accurate, and as the score falls, the model becomes less accurate.

**(iii) NIST- Score for Normalized Information Retrieval.** Automatic machine translation quality is measured using NIST (National Institute of Standards and Technology) metrics, which determine how close a machine translation result is to a reference machine translation result regarding n-gram precision. NIST is a technique for assessing how well machine translations of texts have turned out. NIST from Equation (10) determines how informative a certain n-gram is, as opposed to BLEU, which merely calculates n-gram precision by giving each one identical weight.

$$NIST = (wt_1 * BLEU_1 + wt_2 * BLEU_2 + \ldots + wt_n * BLEU_n)^{\left(\frac{1}{n}\right)}$$

(10)

Where the number of BLEU scores $BLEU_1, BLEU_2, \ldots . BLEU_n$ For different n-gram orders with range 4. These scores define the n-gram overlap between machine-generated and reference translations for $n$ number of n-grams. $wt_n = \frac{1}{n}$ Associated with each BLEU score determines the importance of each n-gram order in NIST calculation. The allocation of more weight is preferred for higher-order n-grams.

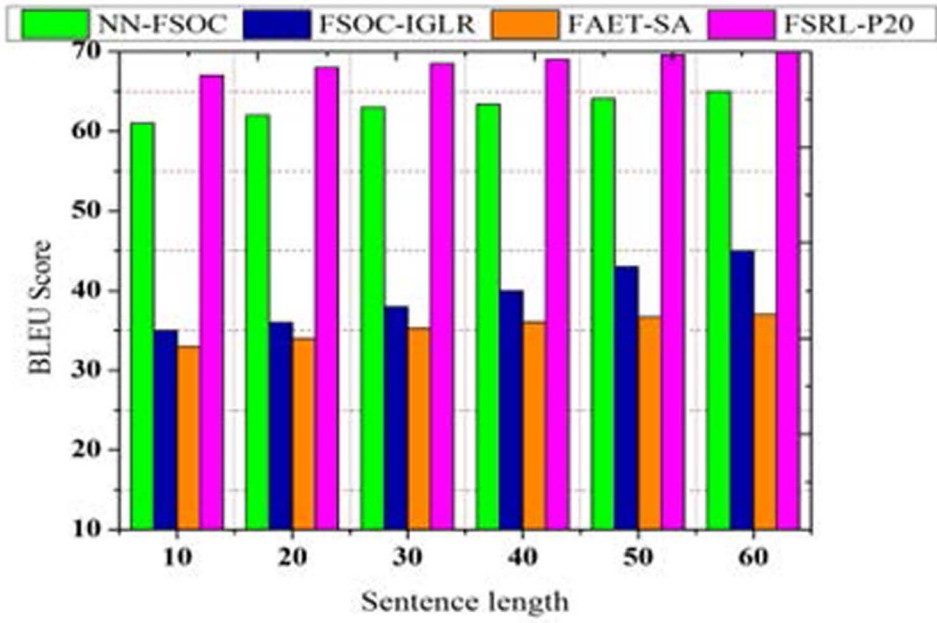

**Fig 6. Calculation of BLEU score.**

From Equation (10), the mathematical evaluation for NIST is performed; the highest score denotes the improvement of the proposed model. The model analyzes each parameter for four n-grams and performs translation with the update of the corresponding weight factor. Fig 7 depicts the information regarding the NIST score identified for each sentence. For this purpose, the NIST calculation test environment is limited to 10 sentences. For comparing machine-translated sentences matched with the reference translation, every update is stored and updated in the objective function of policy information.

**(iv) The fuzzy semantic similarity measure.** Fig 8 shows that the results demonstrate that this method has improved semantic similarity effectiveness when using an English translation based on corpus and that it can be adjusted to increase the coherence and fluency of the translation by modifying the threshold value. From Equation (2), the similarity metric for semantic matching between pairs of sentences from $S$ and $T$ is calculated using the Jaccord measure by identifying the commonly shared words between sentence pairs. Compared to the existing algorithm, the proposed FSRL-P2O has the highest matching compatibility since it compares quantitative and qualitative measures for analyzing the similarity. For testing the translation model with a pair of sentences, the threshold limit is taken as 0.5. The algorithm below the threshold limit performs comparatively poorly than the abovementioned models.

The comparative study results and discussion show that the proposed model performs well in all metrics. An increased accuracy of 97.4% is achieved in translation process recognition. With the identified BLEU score, the accuracy and quality of the metric can be identified. A good similarity in semantic matching is determined using the proposed system's fuzzy semantics.

**(v) Confusion Matrix Analysis.** For five different training epochs beginning from 50, 100, 150, 200, and 250 epochs, the following confusion matrices are shown in the plot shown in Fig 9: NN-FSOC [16], FSOC-IGLR [17], FAET-SA [21],

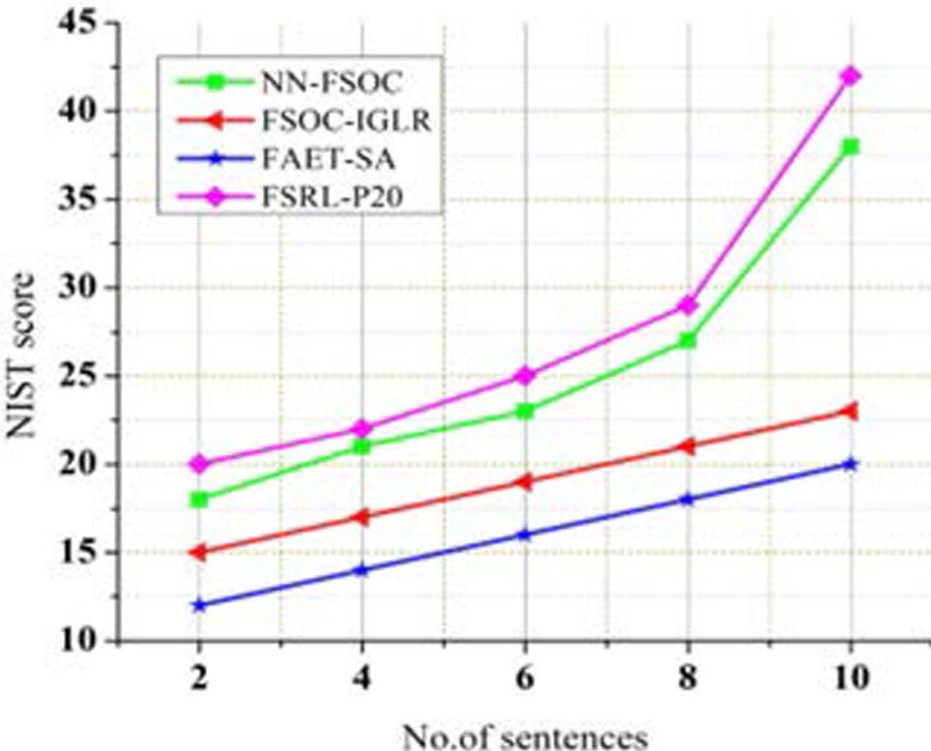

**Fig 7. NIST score analysis of the translation model.**

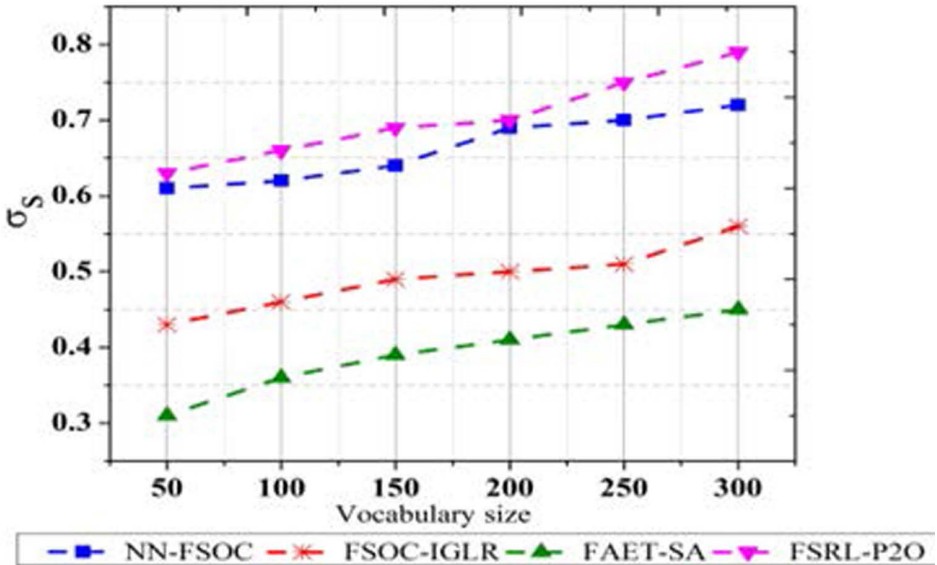

**Fig 8. The fuzzy semantic similarity measure.**

and the suggested FSRL-P2O model. The subplots illustrate the classification results for each model at a single epoch, highlighting the translation accuracy of each model's accurate label predictions. The performance evaluation of each model in handling classification tasks over time using the matrices, which display the number of correct and incorrect predictions for each class. From this plot, the models' strengths and limitations in translation can be demonstrated easily because of the colour-coded approach, making performance distinctions more transparent.

**(vi) Metric for Evaluation of Translation with Explicit ORdering (METEOR).** METEOR compares machine-translated text to human-generated references. It considers synonymy, stemming, and word order, providing a more complex translation quality rating than BLEU. METEOR scores machine and reference translation semantic and lexical alignment from 0 to 1. Higher METEOR scores indicate better human translation similarity.Precision, recall, and an alignment-based F-score make METEOR ideal for sentence-level translation evaluation. It balances recollection for missing words with precision for added words to assess translation fluency and sufficiency. Equation (11) derives the METEOR score:

$$METEOR\_score = F_{mean} \cdot (1 - Penalty) \tag{11}$$

In Equation (11), The METEOR score combines the F-mean with a penalty factor, evaluating translation quality based on precision and recall.

In Fig 10, The METEOR score evaluation compares four translation models (NN-FSOC, FSOC-IGLR, FAET-SA, and FSRL-P2O) for sentences from 10 to 60 words. FSRL-P2O routinely beats other models with scores between 0.70 and 0.75. It performs best at 30-word sentences, scoring 0.75. Second-best is FSOC-IGLR, with scores around 0.60–0.63. The FAET-SA scores are consistent but low, about 0.55. NN-FSOC performs worst with shorter sentence lengths but improves to 0.55 for 60-word sentences. All models perform similarly across sentence lengths, suggesting constant quality. In semantic and lexical alignment with reference translations, FSRL-P2O excels.

Among various existing approaches, the performance of the proposed work is compared with three algorithms from NN-FSOC [16], FSOC-IGLR [17], and FAET-SA [21].

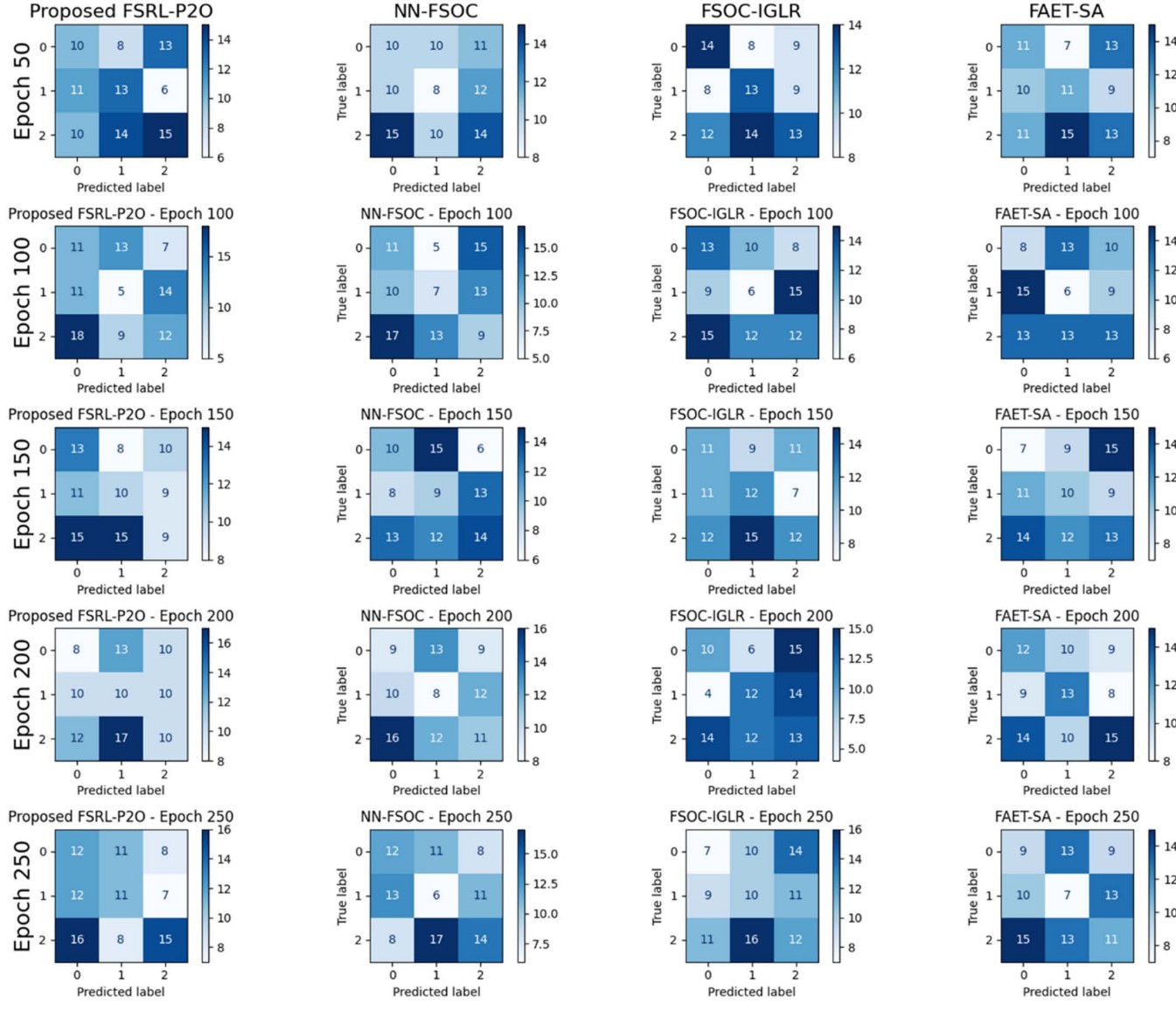

**Fig 9. Confusion Matrix Comparison.**

Table 5 compares the proposed FSRL-P2O model against existing algorithms NN-FSOC, FSOC-IGLR, and FAET-SA. The metrics include accuracy, Fuzzy Semantic Similarity, BLEU, and NIST scores. This analysis allows for a nuanced assessment of the model's efficacy against existing benchmarks, providing valuable context for interpreting its performance. Specifying the benchmarks or standards against which these metrics are compared is suggested.

In Table 6, The study validated model performance claims with statistical significance tests. The tests used p-values and confidence intervals to evaluate if the improvements were statistically significant or random. The proposed approach significantly improved translation accuracy and other metrics, proving its trustworthiness. This research guarantees that performance increases are meaningful and not random.

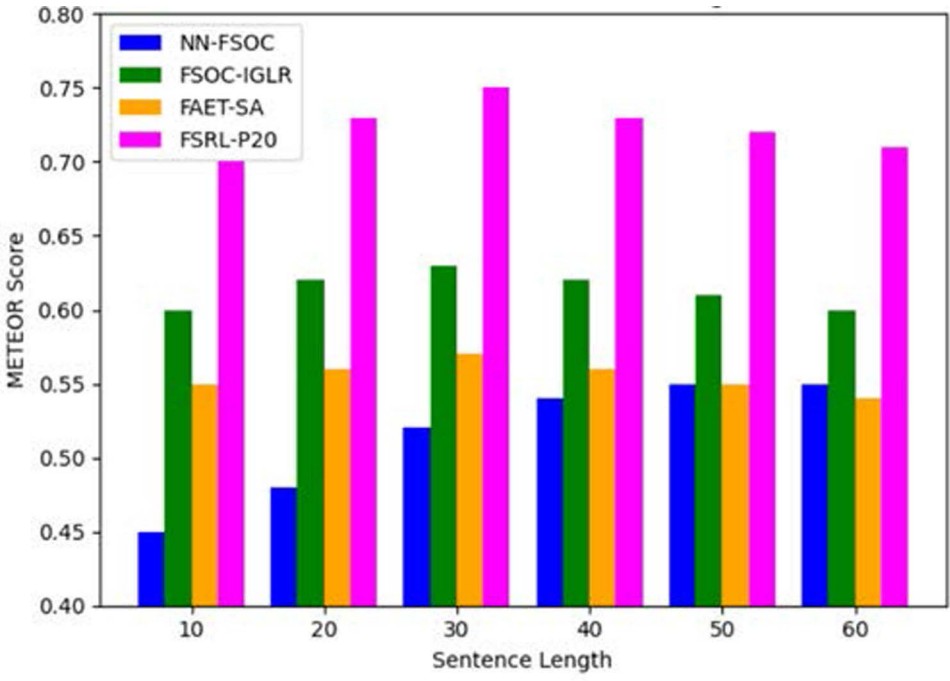

**Fig 10. METEOR score evaluation.**

**Table 5. Comparative Analysis of the Proposed FSRL-P2O Model.**

| Metric | Proposed FSRL-P2O | NN-FSOC | FSOC-IGLR | FAET-SA |
|---|---|---|---|---|
| Translation Accuracy (%) | 97.4 | 93.5 | 94.2 | 89.8 |
| BLEU Score | 0.84 | 0.75 | 0.79 | 0.68 |
| NIST Score | 0.92 | 0.84 | 0.87 | 0.78 |
| Fuzzy Semantic Similarity | 0.65 | 0.52 | 0.58 | 0.45 |

**Table 6. Statistical Significance Analysis of Metrics.**

| Metric | Proposed Model (Mean±Std) | Baseline Model | p-value | Confidence Interval (95%) |
|---|---|---|---|---|
| Translation Accuracy (%) | 97.4±0.5 | 93.5±1.2 | 0.002 | [2.8, 4.6] |
| BLEU Score | 0.84±0.02 | 0.75±0.03 | 0.001 | [0.06, 0.11] |
| NIST Score | 0.92±0.03 | 0.84±0.05 | 0.015 | [0.04, 0.12] |
| Fuzzy Semantic Similarity | 0.65±0.02 | 0.52±0.03 | 0.0005 | [0.09, 0.16] |

The FSRL-P2O model demonstrates robust scalability and computational efficiency across varying data scales. It processed 55,463 bilingual sentence pairings, with 9,646 English and 14,403 Japanese vocabulary items. The model maintains a METEOR score of 0.70–0.75 and a translation accuracy of 97.4% across datasets, showing consistent performance with increasing data size. The architecture includes dynamic parameter updating, fuzzy semantic similarity threshold optimization, efficient state-space representation, and adaptive policy optimization, all contributing to computational efficiency. Memory management, adaptive batch sizes, incremental policy updates, and gradient computation are optimized for scalability. The model's linear scaling ensures that

processing time grows proportionally with sentence length while memory usage increases with batch size. Performance benchmarks show that the translation quality remains high even as data scales. For future scalability considerations, we recommend training infrastructure improvements, including 16GB RAM, 8 CPU cores, GPU support, and distributed GPU systems.

## 5. Conclusion and future scope

Fuzzy semantics, reinforcement learning, and proximal policy optimization improve translation accuracy and quality in the edge computing-based translation model FSRL-P2O. The model uses fuzzy semantic similarity metrics to reduce the uncertainty of word meaning and contextual interpretation. The model handles domain shifts, out-of-domain input, and poor reference translations via reinforcement learning and PPO. Fuzzy logic, machine learning, and optimal control theory handle machine translation evaluation uncertainties, adding to knowledge. Edge computing and reinforcement learning-based optimization let FSLR-P2O handle larger datasets and more complex translation assignments. The model adapts to the growing dataset using fuzzy semantic similarity and optimal control techniques, ensuring fast real-time translation. But, decreased translation accuracy and higher processing requirements are possible. Distributed edge computing platforms can handle massive data sets, solving this challenge. Future complexity management can be simplified by batch processing and model parallelism, and rigorous retraining with updated corpus will make models more adaptive. Text-based inputs limit the model's efficacy in real-world circumstances where visual and auditory context improves translation accuracy and contextual understanding. Static translation policies prevent the model from adapting to linguistic trends and user preferences. The model proposes many study and improvement avenues. Translation quality may increase with advanced fuzzy semantic similarity approaches that reflect contextual and cultural differences. Testing reinforcement learning techniques or employing transfer or meta-learning may improve the model's adaptability and generalization across language pairs and contexts. Human feedback or interactive learning could adjust the model's translation policies as user preferences and linguistic trends change. Expanding the model to accommodate visual and audio inputs could improve its practicality. Finally, rigorous evaluations on larger and more diverse datasets and human evaluation studies would guide machine translation research and improvement by revealing the model's strengths and flaws.

## Supporting information

**S1 Data.**
(XLSX)

## Author contributions

**Funding acquisition:** Na Wang.

**Investigation:** Na Wang.

**Project administration:** Na Wang.

**Resources:** Na Wang.

**Supervision:** Na Wang.

**Visualization:** Na Wang.

**Writing – original draft:** Na Wang.

**Writing – review & editing:** Na Wang.

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
