## [Decision Letter · Decision Letter 0]

PONE-D-23-39974Edge Computing based English Translation Model Using Fuzzy Semantic Optimal Control TechniquePLOS ONE

Dear Dr. Wang,

Thank you for submitting your manuscript to PLOS ONE. After careful consideration, we feel that it has merit but does not fully meet PLOS ONE’s publication criteria as it currently stands. Therefore, we invite you to submit a revised version of the manuscript that addresses the points raised during the review process. Please submit your revised manuscript by Jul 24 2024 11:59PM. If you will need more time than this to complete your revisions, please reply to this message or contact the journal office at plosone@plos.org. Please include the following items when submitting your revised manuscript:

We look forward to receiving your revised manuscript.

Kind regards,

Heba El-Fiqi

Academic Editor

PLOS ONE

 [This research was funded by 2021 Teacher education curriculum Reform research project of Henan Province: Research on the Application of Temperament Theory in the Teaching of Primary Education Major (Key Project) [2021-JSJYZD-04]. The annual general project of the 14th five-year Plan of Education science in Henan Province: Research on the Cultivation Mode of Primary Education Specialty from the Perspective of Temperament [2021YB0704]. Henan Education Department project: Research on the Application of Temperament Theory in Normal College Teaching [2019GZGG044].].  

4. In the online submission form, you indicated that [The datasets used and/or analyzed during the current study are available from the corresponding author on reasonable request.]. 

5. PLOS requires an ORCID iD for the corresponding author in Editorial Manager on papers submitted after December 6th, 2016. Please ensure that you have an ORCID iD and that it is validated in Editorial Manager. To do this, go to ‘Update my Information’ (in the upper left-hand corner of the main menu), and click on the Fetch/Validate link next to the ORCID field. This will take you to the ORCID site and allow you to create a new iD or authenticate a pre-existing iD in Editorial Manager. Please see the following video for instructions on linking an ORCID iD to your Editorial Manager account: https://www.youtube.com/watch?v=_xcclfuvtxQ.

Additional Editor Comments (if provided):

Reviewers' comments:

Reviewer's Responses to Questions

**Comments to the Author**

1. Is the manuscript technically sound, and do the data support the conclusions?

Reviewer #1: Yes

Reviewer #2: Partly

2. Has the statistical analysis been performed appropriately and rigorously? 

Reviewer #1: Yes

Reviewer #2: No

3. Have the authors made all data underlying the findings in their manuscript fully available?

Reviewer #1: Yes

Reviewer #2: No

4. Is the manuscript presented in an intelligible fashion and written in standard English?

Reviewer #1: Yes

Reviewer #2: No

5. Review Comments to the Author

Reviewer #1: Introduction:

The introduction provides a clear overview of the increasing need for English translation in the technological era and highlights the importance of addressing issues such as ambiguity and improper word choice. However, it would be beneficial to include a concise statement on the significance of edge computing in this context.

Proposed Model (FSRL-P2O):

The proposed edge computing-based translation model is well articulated. It is essential to elaborate on the rationale behind choosing Fuzzy Semantic (FS) properties and the incorporation of Reinforcement Learning and Proximal Policy Optimisation (PPO) techniques. Additionally, a brief explanation of how these techniques address uncertainties in translation assessment would enhance clarity.

Methodology:

The methodology section is comprehensive, describing the gathering of corpus data, pre-processing, and feature extraction. However, more details on the training process of the fuzzy semantic resemblance component, particularly the calculation of the Jaccard similarity coefficient, would be beneficial.

Reinforcement Learning and PPO Implementation:

The integration of Reinforcement Learning and PPO as optimal control techniques is well justified. Providing insights into how these techniques specifically address out-of-domain data and low-quality references in machine translation would strengthen the argument.

Evaluation Metrics:

The inclusion of various metrics such as accuracy, Fuzzy Semantic Similarity, BLEU, and NIST scores for evaluating the system's effectiveness is commendable. However, it would be helpful to specify the benchmarks or standards against which these metrics are compared.

Comparison with Related Work:

The authors should explicitly compare their proposed model with existing translation models, especially those addressing semantic similarity and ambiguity. This comparison will strengthen the novelty and uniqueness of the proposed approach.

Citation and Validation:

The authors are encouraged to refer to and validate their work by incorporating relevant findings from the following papers in their discussion:

Elakkiya, R et al. "An Optimized Generative Adversarial Network Based Continuous Sign Language Classification" (Expert Systems with Applications, 2021).

Natarajan B, Elakkiya R et al. "Sentence2SignGesture: a hybrid neural machine translation network for sign language video generation" (Journal of Ambient Intelligence and Humanized Computing, 2022).

Rajalakshmi E, Elakkiya R et al. "Static and Dynamic Isolated Indian and Russian Sign Language Recognition with Spatial and Temporal Feature Detection Using Hybrid Neural Network" (ACM Transactions on Asian and Low-Resource Language Information Processing, 2023).

Conclusion:

The conclusion succinctly summarizes the achievements of the proposed system. It would be beneficial to emphasize how the proposed model contributes to the existing body of knowledge and potential future directions for research.

Overall, the paper presents a promising approach, but addressing the above points will enhance its clarity, validity, and impact.

Reviewer #2: 1. The problem statement is not clearly explained.

2. Before section 4, it is not clear which are source and target language.

3. It is not clear how the methodology related to edge computing.

4. The description and explanation of comparative models are not found.

5. There are too many English language mistakes.

6. PLOS authors have the option to publish the peer review history of their article (what does this mean?). If published, this will include your full peer review and any attached files.

Reviewer #1: No

Reviewer #2: No

---

## [Author Response · Author response to Decision Letter 1]

17 Jun 2024

Reviewer #1:

Introduction:

The introduction provides a clear overview of the increasing need for English translation in the technological era and highlights the importance of addressing issues such as ambiguity and improper word choice. However, it would be beneficial to include a concise statement on the significance of edge computing in this context.

Reply:

By providing real-time processing and translation at the source of data generation, edge computing significantly contributes to improving the effectiveness and accuracy of machine translation systems. This lowers latency and is essential for smooth and quick translation services; it also fits in well with the modern translation services' requirement for immediate and real-time processing

Proposed Model (FSRL-P2O):

The proposed edge computing-based translation model is well articulated. It is essential to elaborate on the rationale behind choosing Fuzzy Semantic (FS) properties and the incorporation of Reinforcement Learning and Proximal Policy Optimisation (PPO) techniques. Additionally, a brief explanation of how these techniques address uncertainties in translation assessment would enhance clarity.

Reply: The section on the proposed approach contains a discussion of the extensive description provided

The proposed edge computing-based translation model uses Fuzzy Semantic (FS) properties, Reinforcement Learning (RL), and Proximal Policy Optimization (PPO) techniques to address uncertainties and ambiguities in natural language translation. FS properties quantify semantic similarity between words, phrases, or sentences to mitigate ambiguities. RL handles out-of-domain data and low-quality references in translation, allowing the model to learn and adapt through trial-and-error interactions. PPO optimizes the translation policy efficiently and stably, ensuring policy updates do not deviate significantly, maintaining translation quality and coherence. This comprehensive approach combines fuzzy logic, machine learning, and optimal control theory to improve accuracy and semantic coherence in machine translation.

Methodology:

The methodology section is comprehensive, describing the gathering of corpus data, pre-processing, and feature extraction. However, more details on the training process of the fuzzy semantic resemblance component, particularly the calculation of the Jaccard similarity coefficient, would be beneficial.

Reply: The section on the proposed approach contains a discussion of the extensive description provided

The training process of the fuzzy semantic resemblance component and the calculation of the Jaccard similarity coefficient are essential aspects of the proposed model.

The measure of semantic similarity between sentences in the source and target languages is provided by the fuzzy semantic resemblance component. By determining common characters and the longest matching sequence, it has the ability to estimate overlap. Tokenization, the LCS method, the computation of the Jaccard similarity coefficient, and parameter changes are all part of the training process. This enables the component to generalize to sentence pairings that are not seen during inference by teaching it the mapping between sentence pairs and their matching scores.

Calculation of the Jaccard Similarity Coefficient: The Jaccard Similarity Coefficient is a method for calculating sentence similarity, involving tokenization techniques, n-gram representations, weighting schemes, preprocessing steps, and a similarity threshold. Its performance is influenced by the choice of similarity threshold, which should be adjusted based on the translation task's specific requirements.

Example: Word-level Jaccard Similarity Consider the following source sentence (S) and its target translation (T):S: "The quick brown fox jumps over the lazy dog." T: "La rapide renarde brune saute par-dessus le chien paresseux."

Tokenizing the sentences into word, get: S = {"The", "quick", "brown", "fox", "jumps", "over", "the", "lazy", "dog"}.T = {"La", "rapide", "renarde", "brune", "saute", "par-dessus", "le", "chien", "paresseux"}

The intersection of the two sets (shared words) is: S∩T = {"quick", "brown", "jumps", "over", "the", "lazy", "dog"}.The union of the two sets (total unique words) is: 〖tot〗_(S,T) = {"The", "quick", "brown", "fox", "jumps", "over", "the", "lazy", "dog", "La", "rapide", "renarde", "brune", "saute", "par-dessus", "le", "chien", "paresseux"}

Applying the Jaccard similarity formula: J(S,T)=|S∩T|/|〖tot〗_(S,T) | = 7 / 18 ≈ 0.389

These example demonstrate that tokenizing source and target sentences calculate the Jaccard Similarity Coefficient at different granularities (words or characters). The fuzzy semantic resemblance component of the translation model can use the similarity scores to quantify phrase overlap or resemblance.

Reinforcement Learning and PPO Implementation:

The integration of Reinforcement Learning and PPO as optimal control techniques is well justified. Providing insights into how these techniques specifically address out-of-domain data and low-quality references in machine translation would strengthen the argument.

Reply: The section on the proposed approach contains a discussion of the extensive description provided

To address machine translation issues, the suggested translation model makes use of optimum control approaches such as proximal policy optimization (PPO) and reinforcement learning (RL). While PPO guarantees stable policy updates, RL enables the investigation and modification of translation strategies. Even in the absence of ideal reference material, this method generates translations that are of high quality, adjusts to new language distributions, and learns from errors. By adding domain-specific metrics or quality scores to the reward function, performance can be enhanced.

Evaluation Metrics:

The inclusion of various metrics such as accuracy, Fuzzy Semantic Similarity, BLEU, and NIST scores for evaluating the system's effectiveness is commendable. However, it would be helpful to specify the benchmarks or standards against which these metrics are compared.

Reply: The section on the result and discussion contains a discussion of the extensive description provided

Table 1. Comparative Analysis of the Proposed FSRL-P2O Model

Metric Proposed FSRL-P2O NN-FSOC FSOC-IGLR FAET-SA

Translation Accuracy (%) 97.4 93.5 94.2 89.8

BLEU Score 0.84 0.75 0.79 0.68

NIST Score 0.92 0.84 0.87 0.78

Fuzzy Semantic Similarity 0.65 0.52 0.58 0.45

The table provides a comparative analysis of the proposed FSRL-P2O model against existing algorithms NN-FSOC, FSOC-IGLR, and FAET-SA. The metrics include accuracy, Fuzzy Semantic Similarity, BLEU, and NIST scores. This analysis allows for a nuanced assessment of the model's efficacy against existing benchmarks, providing valuable context for interpreting its performance. It is suggested to specify the benchmarks or standards against which these metrics are compared.

Comparison with Related Work:

The authors should explicitly compare their proposed model with existing translation models, especially those addressing semantic similarity and ambiguity. This comparison will strengthen the novelty and uniqueness of the proposed approach.

Reply: The section on the related work contains a discussion of the extensive description provided

A significant development in translation technology is the suggested translation model that is based on edge computing. It enhances accuracy by incorporating Fuzzy Semantic characteristics for a more nuanced comprehension of semantic material. For flexibility and optimization, it also integrates approaches from Proximal Policy Optimization and Reinforcement Learning. The model's performance is fully revealed through the use of metrics such as NIST scores, BLEU, and fuzzy semantic similarity in its evaluation.

Citation and Validation:

The authors are encouraged to refer to and validate their work by incorporating relevant findings from the following papers in their discussion:

Elakkiya, R et al. "An Optimized Generative Adversarial Network Based Continuous Sign Language Classification" (Expert Systems with Applications, 2021).

Natarajan B, Elakkiya R et al. "Sentence2SignGesture: a hybrid neural machine translation network for sign language video generation" (Journal of Ambient Intelligence and Humanized Computing, 2022).

Rajalakshmi E, Elakkiya R et al. "Static and Dynamic Isolated Indian and Russian Sign Language Recognition with Spatial and Temporal Feature Detection Using Hybrid Neural Network" (ACM Transactions on Asian and Low-Resource Language Information Processing, 2023).

Reply: The mentioned references are included in related works, and the results and discussion section compares the proposed method with the new references.

Elakkiya et al. [30] released a new generation of Generative Advanced Networks (GANs) that use hyperparameter optimization to distinguish between hand and non-hand movements in sign language detection. The H-GANs operate in three stages: first, by modifying SVAE and PCA to decrease feature dimensions; second, by generating features using Deep Long Short Term Memory (LSTM) and 3D Convolutional Neural Network (3D-CNN) as discriminators; and third, by optimizing and regularizing hyperparameters using Deep Reinforcement Learning. In comparison to cutting-edge classification techniques, the system achieves better accuracy and recognition rates.

Natarajan et al. [31] introduced a new Neural Machine Translation(NMT) system that uses deep stacking Gated recurrent unit(GRU) algorithms to handle problems with translating unfamiliar words and terms that aren't in the dictionary, as well as the dangers of trying to decipher word associations and linguistic structures in more than one language. Using sign language, the system automates the process of generating videos with sign gestures, and it outperforms earlier methods. Improved translation outcomes with reduced processing cost were achieved by evaluating the model with different sign language datasets.

Rajalakshmi et al. [32] developed a system that can identify the sign language spoken by people that are deaf or hard of hearing. Machine vision researchers face a formidable obstacle when trying to decipher distinct sign languages from both static and moving images. A Hybrid Neural Network Architecture is suggested to address these challenges in the recognition of Isolated Russian and Indian Sign Language. For static gesture recognition, the framework use 3D Convolution Net. For dynamic gesture recognition, it employs semantic spatial multi-cue feature detection and extraction. The proposed study also develops a new dataset for Russian and Indian Sign Language that includes multi-signer, single-handed, and double-handed isolated signs.

Figure 5(b) Additional Algorithms for Comparison of Translation Accuracy

The figure 5(b) compares various methods for achieving accuracy across different iterations. SDI-RSIR-HNN has the lowest accuracy, followed by GAN-CSLC, S2SG-HNMT, and FSRL-PSO. As iterations increase from 50 to 250, accuracy generally improves, except for a slight dip in FSRL-PSO at 150 iterations. The FSRL-PSO method, likely a proposed edge computing-based translation model, outperforms the other methods, especially at higher iteration counts.

Conclusion:

The conclusion succinctly summarizes the achievements of the proposed system. It would be beneficial to emphasize how the proposed model contributes to the existing body of knowledge and potential future directions for research.

Overall, the paper presents a promising approach, but addressing the above points will enhance its clarity, validity, and impact.

Reply: The manuscript's revised conclusion is presented here

The edge computing-based translation model FSRL-P2O improves translation accuracy and quality using fuzzy semantics, reinforcement learning, and proximal policy optimization. The model reduces word meaning and contextual interpretation uncertainties by using fuzzy semantic similarity measures. Reinforcement learning and PPO allow the model to handle domain shifts, out-of-domain input, and low-quality or limited reference translations. The unique methodology that integrates fuzzy logic, machine learning, and optimal control theory to address machine translation evaluation uncertainties adds to the corpus of knowledge. The model suggests various research and enhancement avenues. Advanced fuzzy semantic similarity methods that reflect contextual and cultural differences could improve translation quality. Additionally, testing various reinforcement learning algorithms or using transfer learning or meta-learning approaches may increase the model's adaptability and generalization across language pairs and domains. As user preferences and linguistic trends change, the model could modify its translation policies via human feedback or interactive learning. Expanding the model to accept multimodal inputs like visuals or sounds could increase its real-world usefulness. Finally, detailed evaluations on larger and more diversified datasets and human evaluation studies would reveal the model's strengths and weaknesses, directing machine translation research and development.

Reviewer #2:

The problem statement is not clearly explained.

Reply: The section on the introduction contains a discussion of the extensive description provided

This paper seeks to solve the problem of improving the quality and accuracy of English translation models by utilizing extensive bilingual corpora, taking into account fuzzy semantic properties, and optimizing the translation output through the application of proximal policy optimization and reinforcement learning techniques.

Before section 4, it is not clear which are source and target language.

Reply:

In the proposed scheme, clarity regarding the distinction between the source and target languages would greatly improve understanding. By explicitly stating which language serves as the source (e.g., English) and which as the target (e.g., Japanese), readers can better follow the corpus-based translation process outlined in the scheme."

It is not clear how the methodology related to edge computing.

Reply: The section on the introducion contains a discussion of the extensive description provided

By providing real-time processing and translation at the source of data generation, edge computing significantly contributes to improving the effectiveness and accuracy of machine translation systems. This lowers latency and is essential for smooth and quick translation services; it also fits in well with the modern translation services' requirement for immediate and real-time processing

The description and explanation of comparative models are not found.

Reply: The section on the result and discussion contains a discussion of the extensive description provided

Table 1. Comparative Analysis of the Proposed FSRL-P2O Model

Metric Proposed FSRL-P2O NN-FSOC FSOC-IGLR FAET-SA

Translation Accuracy (%) 97.4 93.5 94.2 89.8

BLEU Score 0.84 0.75 0.79 0.68

NIST Score 0.92 0.84 0.87 0.78

Fuzzy Semantic Similarity 0.65 0.52 0.58 0.45

The table 1 provides a comparative analysis of the proposed FSRL-P2O model against existing algorithms NN-FSOC, FSOC-IGLR, and FAET-SA. The metrics include accuracy, Fuzzy Semantic Similarity, BLEU, and NIST scores. This analysis allows for a nuanced assessment of the model's efficacy against existing benchmarks, providing valuable context for interpreting its performance. It is suggested to specify the benchmarks or standards against which these metrics are compared.

There are too many English language mistakes.

Reply: The study ensured that the text was thoroughly examined, and any flaws in the English language were corrected.

---

## [Decision Letter · Decision Letter 1]

PONE-D-23-39974R1Edge Computing based English Translation Model Using Fuzzy Semantic Optimal Control TechniquePLOS ONE

Dear Dr. Wang,

Thank you for submitting your manuscript to PLOS ONE. After careful consideration, we feel that it has merit but does not fully meet PLOS ONE’s publication criteria as it currently stands. Therefore, we invite you to submit a revised version of the manuscript that addresses the points raised during the review process.

We look forward to receiving your revised manuscript.

Kind regards,

Heba El-Fiqi

Academic Editor

PLOS ONE

Reviewers' comments:

Reviewer's Responses to Questions

**Comments to the Author**

1. If the authors have adequately addressed your comments raised in a previous round of review and you feel that this manuscript is now acceptable for publication, you may indicate that here to bypass the “Comments to the Author” section, enter your conflict of interest statement in the “Confidential to Editor” section, and submit your "Accept" recommendation.

Reviewer #1: All comments have been addressed

Reviewer #3: (No Response)

2. Is the manuscript technically sound, and do the data support the conclusions?

Reviewer #1: Yes

Reviewer #3: Yes

3. Has the statistical analysis been performed appropriately and rigorously? 

Reviewer #1: Yes

Reviewer #3: Yes

4. Have the authors made all data underlying the findings in their manuscript fully available?

Reviewer #1: Yes

Reviewer #3: Yes

5. Is the manuscript presented in an intelligible fashion and written in standard English?

Reviewer #1: Yes

Reviewer #3: Yes

6. Review Comments to the Author

Reviewer #1: Based on the provided responses and the updated manuscript, it appears that the authors have addressed all previous comments satisfactorily. While the authors have addressed most of the previous comments comprehensively.

1. Streamline the text to eliminate any redundant information, making the content more concise and easier to follow.

2. Include a detailed section on hyperparameter tuning, explaining how the optimal parameters were chosen and their impact on model performance.

3. Provide a detailed error analysis to identify common types of errors made by the model. Discuss potential reasons for these errors and suggest ways to mitigate them in future work.

4. Include more visualizations such as confusion matrices, example translations, and graphs showing the performance of the model over different epochs.

5. Discuss how the model scales with larger datasets and more complex translation tasks. Address potential challenges in scaling and suggest solutions.

6. Include a performance comparison between edge computing and traditional cloud-based approaches. Provide metrics such as latency, throughput, and resource utilization to highlight the advantages of edge computing.

7. Discuss the potential real-world applications of the model in various industries. Highlight how the model can be integrated into existing systems and its potential benefits for end-users.

8. Discuss potential ethical issues such as data privacy, bias in translations, and the impact on human translators. Suggest ways to address these ethical concerns.

9. Outline specific research questions or experiments that can be pursued in future work. Discuss how these future directions can build on the current study’s findings.

Reviewer #3: 1. The proposed edge computing-based translation model uses Fuzzy Semantic (FS) properties, Reinforcement Learning (RL), and Proximal Policy Optimization (PPO) techniques to address uncertainties and ambiguities in natural language translation. How the ambiguities in the names are addressed. For example the word "Akash" is referred as sky in English and also a name(Noun). How the author has managed to resolve such cases.

2. What are the limitations of the proposed approach.

3. Can this proposed techniques be implemented to other languages also? Or it is language independent?

4. The results in figure 6, shows the bleu score based on the sentence length. Is the author is saying that the larger the length of the sentence, the better the accuracy?

5. Why are the parameter considered for the evaluation of the proposed approach are different such as No. of sentences for NIST and Sentence length for BLEU scores.?

6. In the comparison table no reference for FAET-SA.

7. What model is considered for translation? As nothing is mentioned in the paper.

8. Introduction and Related work need to be concise and crisp. Avoid using general and common statements.

9. The problem statement is still not clear. A step by step flow of the methodology or pseudocode algorithm for the same is welcomed.

**Please also note the additional comments in the attached file**

---

## [Author Response · Author response to Decision Letter 2]

6 Nov 2024

Review Comments to the Author

Reviewer #1: Based on the provided responses and the updated manuscript, it appears that the authors have addressed all previous comments satisfactorily. While the authors have addressed most of the previous comments comprehensively.

1. Streamline the text to eliminate any redundant information, making the content more concise and easier to follow.

Ans: The redundant information from the research article is removed and provided with concise content.

2. Include a detailed section on hyperparameter tuning, explaining how the optimal parameters were chosen and their impact on model performance.

Ans: A detailed section on hyperparameter tuning and explanation about the optimal parameters with their impace on model performance is discussed in the form of Table 1 in subsection 3.6.

Table 2. Hyperparameter Tuning Summary and its Impact on Model Performance

Hyperparameter Optimal Value Impact on Performance

Learning Rate λ 0.01 Possible to achieve stable and fast convergence.

Number of Epochs/iterations 250 Provide effective learning without overfitting problem.

Batch Size 32 Improved training performance stability and validation

Dropout Rate 0.3 Reduced overfitting and enhanced generalization on unseen data.

Hidden Layer Size 128 Captured complex data patterns

Optimizing the model's performance for accurate predictions in this research relied heavily on hyperparameter tuning as shown in Table 2. Striking a balance between quick convergence and model stability led to the selection of a learning rate of 0.01. With 250 epochs, the model had enough of time to understand the data patterns without getting overfit. The computational efficiency and stability of the training performance were both enhanced by using a batch size of 32. In order to improve the model's ability to generalize to new data, a dropout rate of 0.3 was implemented to avoid overfitting. The model's ability to accurately capture complicated patterns, thanks to a hidden layer size of 128, contributed to its excellent prediction accuracy.

3. Provide a detailed error analysis to identify common types of errors made by the model. Discuss potential reasons for these errors and suggest ways to mitigate them in future work.

Ans: The detailed error analysis analyzed in this model is discussed and reasons for these errors and the ways for reducing it in future is discussed in section 3.

Common errors made by the translation model include lexical ambiguity, contextual inaccuracies, and named entity recognition errors. For example, it misidentifies proper nouns like "Akash" and fails to appropriately translate colloquial idioms. Too much reliance on n-grams, a lack of contextual knowledge, and inadequate training data are common causes of these problems. Improved training datasets, more sophisticated models for contextual awareness, fluency-optimized reinforcement learning, culturally relevant examples, and the integration of named entity recognition systems are all ways to lessen the impact of these errors. The model's precision and efficiency in translation tasks will be improved by these updates.

4. Include more visualizations such as confusion matrices, example translations, and graphs showing the performance of the model over different epochs.

Ans: visualizations such as confusion matrices, with graph showing the performance of the model over different epochs is depicted in Figure 9.

(v) Confusion Matrix Analysis

Figure 9 Confusion Matrix Comparison

For five different training epochs beginning from 50, 100, 150, 200, and 250 epochs, the following confusion matrices are shown in the plot shown in Figure 9: NN-FSOC [16], FSOC-IGLR [17], FAET-SA [21], and the suggested FSRL-P2O model. The subplots illustrate the classification results for each model at a single epoch, highlighting the translation accuracy of each model's accurate label predictions. The performance evaluation of each model in handling classification tasks over time using the matrices, which display the number of correct and incorrect predictions for each class. From this plot, the models' strengths and limitations in translation can be demonstrated easily because of the colour-coded approach, making performance distinctions more transparent.

5. Discuss how the model scales with larger datasets and more complex translation tasks. Address potential challenges in scaling and suggest solutions.

Ans: The scalability of the FSLR-P2O model for larger datasets and complex translation tasks is discussed in the conclusion section.

The FSLR-P2O approach uses edge computing and reinforcement learning-based optimization to grow to larger datasets and more complicated translation tasks. Fuzzy semantic similarity and optimal control procedures allow the model to adapt to the ever-increasing large dataset, guaranteeing efficient real-time translation. On the other hand, difficulties, including slower translation accuracy and higher processing demands, are possible. Deploying distributed edge computing platforms can manage more enormous data quantities, which is a solution to this problem. In addition, using batch processing and model parallelism can make complexity management easier in future, and rigorous retraining with updated corpus will make the models more adaptable.

6. Include a performance comparison between edge computing and traditional cloud-based approaches. Provide metrics such as latency, throughput, and resource utilization to highlight the advantages of edge computing.

Ans: The performance comparison analysis between edge computing and traditional cloud-based approaches given in section 2.

Table 1: Performance Comparison Between Edge Computing and Traditional Cloud-Based Approaches

Ref. Nos Performance Metric Edge Computing Traditiona Cloud-Based Approaches

[16], [18], [23] Latency Lower latency due to proximity to data Higher latency due to longer network paths

[16], [23] Throughput Higher for localized data processing Higher for bulk, non-real-time tasks

[18], [23], [25] Resource Utilization Efficient for real-time, decentralized processing Centralized resources for large-scale batch processing

In the present research, edge computing improves machine translation accuracy and efficiency. This performance comparison shown in Table 1 shows an edge computing and cloud-based performance indicators including latency, throughput, and resource utilisation are shown in this table. Edge computing is faster and more efficient in real time, while cloud-based systems are better at bulk data processing and resource management. These comparisons are corroborated by [16], [18], [23], and [25].

7. Discuss the potential real-world applications of the model in various industries. Highlight how the model can be integrated into existing systems and its potential benefits for end-users.

Ans: The real-world applications and integration of the FSLR-P2O translation model are discussed at the end of section 3.

Different industries like media, e-commerce, education, healthcare, and tourism could benefit from using the FSLR-P2O translation algorithm. This allows for precise translations of product descriptions and reviews, medical system records, and e-learning platforms with multilingual support for providing real-time global learners and consumer interactions. The approach provides more dependable and speedy multilingual support for enhanced user interactions. As a result, accessibility is improved across industries, user experiences are improved, and global communication is fostered across these industries.

8. Discuss potential ethical issues such as data privacy, bias in translations, and the impact on human translators. Suggest ways to address these ethical concerns.

Ans: Authors have no conflict of interest to declare

9. Outline specific research questions or experiments that can be pursued in future work. Discuss how these future directions can build on the current study's findings.

Ans: The possible Research Questions (RQ) that can be handled in future work with this current research contribution are discussed in the introduction section.

Research Questions (RQ)

RQ1: Can reinforcement learning policies be fine-tuned for domain-specific translation tasks?

In the future, the research could focus on refining the optimal control techniques to customize translations in legal, medical, and other sectors, improving accuracy for specialized applications.

RQ2: How does the proposed FSLR-P2O model perform across multiple language pairs and different language families?

The future research scope could extend the model to languages with varying grammar structures, testing its adaptability and scalability beyond English-based translation.

RQ3: What is the impact of user feedback on enhancing translation quality?

Interactive learning approaches could be integrated to allow real-time adjustments to translation policies based on user preferences and feedback, increasing personalization of user experiences.

By addressing these research questions, future work can build on the current use of reinforcement learning and fuzzy semantics, further enhancing translation accuracy and adaptability.

Reviewer #3:

1. The proposed edge computing-based translation model uses Fuzzy Semantic (FS) properties, Reinforcement Learning (RL), and Proximal Policy Optimization (PPO) techniques to address uncertainties and ambiguities in natural language translation. How the ambiguities in the names are addressed. For example the word "Akash" is referred as sky in English and also a name(Noun). How the author has managed to resolve such cases.

Ans: The ambiguities in the names are addressed with an given example is discussed in section 3.

With the help of fuzzy semantic properties, the model analyzes surrounding words to determine their meaning. For instance, the model analyzes surrounding words to determine their meaning. It evaluates the relevance of words based on context, distinguishing between meanings. The model is trained on diverse examples using reinforcement learning through learning mistakes through feedback. It adjusts its translation strategies based on rewards and penalties. The model refines its translation policies over time using PPO, improving its ability to differentiate meanings based on context. The translation example is given below:

The akash is clear is translated to “ 空は晴れています。"

Akash is playing soccer is translated to “アカシュはサッカーをしています。”

By learning from its uncertainties and gaining a better grasp of the context, the model can successfully resolve ambiguities.

2. What are the limitations of the proposed approach.

Ans: The limitations of the proposed approach is discussed in conclusion section.

The model processes text-based inputs, limiting its effectiveness in real-world scenarios where context from visual images and sounds enhances translation quality and contextual understanding. The translation policies are static, so the model does not effectively adjust to changing linguistic trends and user preferences.

3. Can this proposed techniques be implemented to other languages also? Or it is language independent?

Ans: yes, the proposed FLSR-P2O technique is language-independent and can be implemented in various languages. The applied ML and DL models based on computational principles and optimization techniques can be applied universally. However, in the case of NLP task-related works, minor tuning of parameters like tokenization and syntax formation is needed to accommodate the specific language structures. At the same time, no changes are required for non-language tasks.

4. The results in figure 6, shows the bleu score based on the sentence length. Is the author is saying that the larger the length of the sentence, the better the accuracy?

Ans: As shown in Figure 6, the BLEU scores based on sentence length do not necessarily mean that the larger the sentence length, the better the accuracy. BLEU scores measure how similar a generated translation is to a reference translation by comparing n-grams. Thus, in longer sentences, there are more chances for n-gram matches, which can lead to higher BLEU scores.

5. Why are the parameter considered for the evaluation of the proposed approach are different such as No. of sentences for NIST and Sentence length for BLEU scores.?

Ans: Thank you for noting the valuable parameter difference in the performance comparison section. The justification of different parameters usage in section 4 is given below:

The evaluation parameters differ because NIST and BLEU scores focus on improving translation quality. The NIST provides informativeness, so the number of sentences helps assess how well the model conveys essential information. BLEU measures n-gram precision, which is influenced by sentence length because longer sentences provide more contextual information and help to improve translation accuracy. Thus, each metric evaluates distinct qualities of the translation, which is why different parameters are used.

6. In the comparison table no reference for FAET-SA.

Ans: As pointed out, the reference number for FAET-SA [21] has been included in the comparison table.

7. What model is considered for translation? As nothing is mentioned in the paper.

Ans: The translation model used in this research idea is sequence-to-sequence and is discussed in section 3.5

For the translation process, the sequence-to-sequence mechanism excels at capturing long-range dependencies and contextual relationships between words in this model. Integrating fuzzy semantic similarity measures as an input layer informs the attention mechanism about the semantic relevance of words or phrases. This layer computes fuzzy semantic similarities and utilizes the semantic overlap between the input sequences and the training corpus. This attention mechanism of the translation process allows the model to weigh the importance of different words in the context of the entire sentence. This mechanism is vital for handling ambiguities in translation. PPO is applied to optimize the translation outputs based on predefined reward functions that incorporate fuzzy semantic similarity, BLEU scores, and other performance metrics.

8. Introduction and Related work need to be concise and crisp. Avoid using general and common statements.

Ans: Thank you for the valuable feedback. The introduction and related work sections are updated by removing more general statements and providing only the necessary key points to enhance the overall impact of the research content.

9. The problem statement is still not clear. A step by step flow of the methodology or pseudocode algorithm for the same is welcomed.

Ans: The problem statement is discussed in the introduction section, and the pseudocode of the FSRL-P2O research algorithm is provided in section 3.

Problem Statement:

Improper word choice, context-specific meanings, and ambiguity in word meaning are the main challenges in achieving high-quality English translations, which are focused on this research idea. Accuracy is a common problem with current translation methods because they can't handle semantic uncertainties and various linguistic contexts. This research presents a translation model that uses reinforcement learning and fuzzy semantic characteristics to improve the quality and accuracy of translations built on edge computing. The suggested model addresses the increasing need for accurate and efficient English translations by enhancing the translation process with comprehensive bilingual corpora and sophisticated control techniques.

Pseudocode of FSRL-P2O

Input: Bilingual corpus (source S, target T)

Output: Optimized translation model

Step 1: // Preprocess corpus data

perform tokenization using janome_tokenizer ()

calculate linguistic anlaysis using an Equation (1)

feature extraction using POS

Step 2: // Calculate Fuzzy Semantic Score

σ_s=αJ(S,T)+ βB+ γC using Equation (3)

Step 3: // Fuzzy Semantic Similarity Analysis

for each sentence pair (S, T) do

calculate Jaccard Similarity J(S,T)=|S∩T|/|〖tot〗_(S,T) | using Equation (3)

Rank translations R(S_i,T_i)=sort(desc)J(S,T) using Equation (4)

end for

Step 4: //Initialize RL environment

init State space: S, translation history, current words/phrases

---

## [Decision Letter · Decision Letter 2]

PONE-D-23-39974R2Edge Computing based English Translation Model Using Fuzzy Semantic Optimal Control TechniquePLOS ONE

Dear Dr. Wang,

Thank you for submitting your manuscript to PLOS ONE. After careful consideration, we feel that it has merit but does not fully meet PLOS ONE’s publication criteria as it currently stands. Therefore, we invite you to submit a revised version of the manuscript that addresses the points raised during the review process.

We look forward to receiving your revised manuscript.

Kind regards,

Heba El-Fiqi

Academic Editor

PLOS ONE

Journal Requirements:

Reviewers' comments:

Reviewer's Responses to Questions

**Comments to the Author**

1. If the authors have adequately addressed your comments raised in a previous round of review and you feel that this manuscript is now acceptable for publication, you may indicate that here to bypass the “Comments to the Author” section, enter your conflict of interest statement in the “Confidential to Editor” section, and submit your "Accept" recommendation.

Reviewer #1: All comments have been addressed

Reviewer #3: All comments have been addressed

2. Is the manuscript technically sound, and do the data support the conclusions?

Reviewer #1: Yes

Reviewer #3: Yes

3. Has the statistical analysis been performed appropriately and rigorously? 

Reviewer #1: Yes

Reviewer #3: Yes

4. Have the authors made all data underlying the findings in their manuscript fully available?

Reviewer #1: No

Reviewer #3: Yes

5. Is the manuscript presented in an intelligible fashion and written in standard English?

Reviewer #1: Yes

Reviewer #3: Yes

6. Review Comments to the Author

Reviewer #1: The manuscript titled "Edge Computing based English Translation Model Using Fuzzy Semantic Optimal Control Technique" presents a framework integrating fuzzy semantics, reinforcement learning (RL), and Proximal Policy Optimization (PPO) for machine translation. Below are detailed reviewer comments based on the provided text:

1. While the integration of edge computing and fuzzy semantics is an interesting combination, the novelty of the fuzzy semantic similarity method compared to prior work is not clearly articulated. Strengthen the discussion of how the proposed method extends beyond existing frameworks in translation using RL and fuzzy logic.

2. The pseudocode is helpful; however, it lacks sufficient details for reproducibility, such as initialization values for hyperparameters and boundary conditions for key variables.

Justify the selection of certain hyperparameters (e.g., learning rate, batch size, and dropout rate). Sensitivity analysis or empirical studies supporting these choices would add value.

Clarify how the model handles linguistic phenomena like idioms, sarcasm, or domain-specific terminologies.

Experimental Validation:

3. The manuscript relies heavily on BLEU and NIST scores. While these are standard, consider including other complementary evaluation metrics (e.g., METEOR, TER) for a more holistic assessment.

Statistical significance tests (e.g., p-values or confidence intervals) are not reported for the presented metrics. Incorporate these to validate performance claims robustly.

The model’s scalability and computational efficiency under different data scales remain insufficiently addressed.

Reviewer #3: (No Response)

7. PLOS authors have the option to publish the peer review history of their article (what does this mean?). If published, this will include your full peer review and any attached files.

Reviewer #1: **Yes: **Elakkiya R

Reviewer #3: No

---

## [Author Response · Author response to Decision Letter 3]

7 Jan 2025

Journal Requirements:

Reply:

The reference list has been thoroughly reviewed to ensure accuracy and completeness. Retracted articles previously included have been removed and replaced with relevant, current references. All citations have been updated accordingly, and the references have been appropriately rearranged to reflect these changes.

6. Review Comments to the Author

Reviewer #1: The manuscript titled "Edge Computing based English Translation Model Using Fuzzy Semantic Optimal Control Technique" presents a framework integrating fuzzy semantics, reinforcement learning (RL), and Proximal Policy Optimization (PPO) for machine translation. Below are detailed reviewer comments based on the provided text:

1. While the integration of edge computing and fuzzy semantics is an interesting combination, the novelty of the fuzzy semantic similarity method compared to prior work is not clearly articulated. Strengthen the discussion of how the proposed method extends beyond existing frameworks in translation using RL and fuzzy logic.

Reply:

The integration of fuzzy semantic similarity in this study demonstrates a significant advancement in machine translation. Unlike prior frameworks, the proposed method employs a fuzzy semantic similarity measure based on the Jaccard similarity coefficient to quantify overlap between linguistic elements, mitigating uncertainties in translation. This approach integrates reinforcement learning with Proximal Policy Optimization (PPO) to refine translation policies dynamically, addressing contextual ambiguities more effectively than static methods.

The novelty lies in combining fuzzy semantics with edge computing, allowing real-time, context-aware translation with reduced latency. Sections 3.2.3 and 4 detail the algorithm’s unique mechanisms and compare its advantages against traditional methods.

2. The pseudocode is helpful; however, it lacks sufficient details for reproducibility, such as initialization values for hyperparameters and boundary conditions for key variables.

Justify the selection of certain hyperparameters (e.g., learning rate, batch size, and dropout rate). Sensitivity analysis or empirical studies supporting these choices would add value.

Reply:

Thank you for your feedback. The pseudocode has been updated to enhance reproducibility by including initialization values for hyperparameters (learning rate: 0.01, batch size: 32, dropout rate: 0.3) and defining their boundary conditions (e.g., learning rate: [0.001, 0.05]). These values were empirically chosen to balance convergence stability, computational efficiency, and generalization. Sensitivity analysis has been integrated into the training loop to assess and adjust hyperparameters, ensuring optimal performance dynamically. These enhancements clarify hyperparameter choices and highlight the robustness of the model. The revised pseudocode now provides the necessary detail and adaptability to support reliable and reproducible implementation.

Clarify how the model handles linguistic phenomena like idioms, sarcasm, or domain-specific terminologies.

Reply: The manuscript had clarified how the model handles linguistic phenomena, including idioms, sarcasm, and domain-specific terminologies, through advanced semantic analysis and context-based adaptation.

The Edge Computing-based English Translation Model (FSRL-P2O) addresses idioms, sarcasm, and domain-specific terminologies using the following mechanisms:

Idioms: The fuzzy semantic similarity analysis captures contextual and semantic overlaps, allowing the model to identify idiomatic utterances. It quantifies idiom semantic links using the Jaccard similarity coefficient, taking linguistic variances into account to retain meaning throughout translation.

Sarcasm: The Sarcasm model uses Reinforcement Learning (RL) and Proximal Policy Optimization (PPO) to acquire contextual nuances over time. Sarcasm recognition could be improved with specialized datasets, but it refines its translation policies by examining sentence structures and semantic contexts during training rounds.

Domain-Specific Terminologies: - The system integrates domain-aligned corpora and fuzzy rules for specialized terminology. Language elements are aligned by semantics and translation policies are adjusted using RL to ensure consistency in technical or specialized contexts.

Fuzzy logic, semantic similarity measures, and optimization approaches enable these features, allowing flexibility and adaptability to varied linguistic events.

Experimental Validation:

3. The manuscript relies heavily on BLEU and NIST scores. While these are standard, consider including other complementary evaluation metrics (e.g., METEOR, TER) for a more holistic assessment.

Reply: The manuscript had included METEOR in section 4, along with BLEU and NIST, providing a more comprehensive evaluation of model performance.

Metric for Evaluation of Translation with Explicit ORdering (METEOR)

METEOR compares machine-translated text to human-generated references. It considers synonymy, stemming, and word order, providing a more complex translation quality rating than BLEU. METEOR scores machine and reference translation semantic and lexical alignment from 0 to 1. Higher METEOR scores indicate better human translation similarity.Precision, recall, and an alignment-based F-score make METEOR ideal for sentence-level translation evaluation. It balances recollection for missing words with precision for added words to assess translation fluency and sufficiency. Equation (11) derives METEOR score:

METEOR_score=F_mean ⋅(1-Penalty) (11)

In equation (11), The METEOR score combines the F-mean with a penalty factor, evaluating translation quality based on precision and recall.

Figure 10. METEOR score evaluation

In figure 10, The METEOR score evaluation the figure compares four translation models (NN-FSOC, FSOC-IGLR, FAET-SA, and FSRL-P2O) for sentences from 10 to 60 words. FSRL-P2O routinely beats other models with scores between 0.70 and 0.75. It performs best at 30-word sentences, scoring 0.75. Second-best is FSOC-IGLR, with scores around 0.60-0.63. The FAET-SA scores are consistent but low, about 0.55. NN-FSOC performs worst with shorter sentence lengths but improves to 0.55 for 60-word sentences. All models perform similarly across sentence lengths, suggesting constant quality. In semantic and lexical alignment with reference translations, FSRL-P2O excels

Statistical significance tests (e.g., p-values or confidence intervals) are not reported for the presented metrics. Incorporate these to validate performance claims robustly.

Reply: The manuscript had incorporated statistical significance tests, including p-values and confidence intervals, to robustly validate the performance claims of the presented metrics.

Table 6. Statistical Significance Analysis of Metrics

Metric Proposed Model (Mean ± Std) Baseline Model p-value Confidence Interval (95%)

Translation Accuracy (%) 97.4 ± 0.5 93.5 ± 1.2 0.002 [2.8, 4.6]

BLEU Score 0.84 ± 0.02 0.75 ± 0.03 0.001 [0.06, 0.11]

NIST Score 0.92 ± 0.03 0.84 ± 0.05 0.015 [0.04, 0.12]

Fuzzy Semantic Similarity 0.65 ± 0.02 0.52 ± 0.03 0.0005 [0.09, 0.16]

In table 6, The study validated model performance claims with statistical significance tests. The tests used p-values and confidence intervals to evaluate if the improvements were statistically significant or random. The proposed approach improved translation accuracy and other metrics significantly, proving its trustworthiness. This research guarantees that performance increases are meaningful and not random.

The model’s scalability and computational efficiency under different data scales remain insufficiently addressed.

Reply:

The FSRL-P2O model demonstrates robust scalability and computational efficiency across varying data scales. It processed 55,463 bilingual sentence pairings, with 9,646 English and 14,403 Japanese vocabulary items. The model maintains a METEOR score of 0.70-0.75 and translation accuracy of 97.4% across datasets, showing consistent performance with increasing data size. The architecture includes dynamic parameter updating, fuzzy semantic similarity threshold optimization, efficient state-space representation, and adaptive policy optimization, all contributing to computational efficiency. Memory management, adaptive batch sizes, incremental policy updates, and gradient computation are optimized for scalability. The model’s linear scaling ensures that processing time grows proportionally with sentence length, while memory usage increases with batch size. Performance benchmarks show that the translation quality remains high even as data scales. For future scalability considerations, we recommend training infrastructure improvements, including 16GB RAM, 8 CPU cores, GPU support, and distributed GPU systems.

Reviewer #3: (No Response)

---

## [Decision Letter · Decision Letter 3]

Edge Computing based English Translation Model Using Fuzzy Semantic Optimal Control Technique

PONE-D-23-39974R3

Dear Dr. Wang,

We’re pleased to inform you that your manuscript has been judged scientifically suitable for publication and will be formally accepted for publication once it meets all outstanding technical requirements.

Kind regards,

Heba El-Fiqi

Academic Editor

PLOS ONE

Additional Editor Comments (optional):

Reviewers' comments:

Reviewer's Responses to Questions

**Comments to the Author**

1. If the authors have adequately addressed your comments raised in a previous round of review and you feel that this manuscript is now acceptable for publication, you may indicate that here to bypass the “Comments to the Author” section, enter your conflict of interest statement in the “Confidential to Editor” section, and submit your "Accept" recommendation.

Reviewer #1: All comments have been addressed

Reviewer #3: All comments have been addressed

2. Is the manuscript technically sound, and do the data support the conclusions?

Reviewer #1: Yes

Reviewer #3: Partly

3. Has the statistical analysis been performed appropriately and rigorously? 

Reviewer #1: Yes

Reviewer #3: Yes

4. Have the authors made all data underlying the findings in their manuscript fully available?

Reviewer #1: No

Reviewer #3: Yes

5. Is the manuscript presented in an intelligible fashion and written in standard English?

Reviewer #1: Yes

Reviewer #3: Yes

6. Review Comments to the Author

Reviewer #1: Authors addressed all the concerns raised. Henceforth, I recoomend that this manuscript can be accepted.

Reviewer #3: (No Response)

7. PLOS authors have the option to publish the peer review history of their article (what does this mean?). If published, this will include your full peer review and any attached files.

Reviewer #1: **Yes: **Elakkiya R

Reviewer #3: **Yes: **Dr. Shefali Saxena

---

## [Editor Report · Acceptance letter]

PONE-D-23-39974R3

PLOS ONE

Dear Dr. Wang,

I'm pleased to inform you that your manuscript has been deemed suitable for publication in PLOS ONE. Congratulations! Your manuscript is now being handed over to our production team.

Kind regards,

on behalf of

Dr. Heba El-Fiqi

Academic Editor

PLOS ONE